# Efficient LiDAR Reflectance Compression via Scanning Serialization

Jiahao Zhu [* 1]   Kang You [* 2]   Dandan Ding [1]   Zhan Ma [2]

## Abstract

Reflectance attributes in LiDAR point clouds provide essential information for downstream tasks but remain underexplored in neural compression methods. To address this, we introduce SerLiC, a serialization-based neural compression framework to fully exploit the intrinsic characteristics of LiDAR reflectance. SerLiC first transforms 3D LiDAR point clouds into 1D sequences via scan-order serialization, offering a device-centric perspective for reflectance analysis. Each point is then tokenized into a contextual representation comprising its sensor scanning index, radial distance, and prior reflectance, for effective dependencies exploration. For efficient sequential modeling, Mamba is incorporated with a dual parallelization scheme, enabling simultaneous autoregressive dependency capture and fast processing. Extensive experiments demonstrate that SerLiC attains over $2\times$ volume reduction against the original reflectance data, outperforming the state-of-the-art method by up to 22% reduction of compressed bits while using only 2% of its parameters. Moreover, a lightweight version of SerLiC achieves $\geq 10$ fps (frames per second) with just 111K parameters, which is attractive for real-world applications.

## 1. Introduction

Over the past two decades, the light detection and ranging (LiDAR) sensors have become indispensable components in a diverse array of applications, including autonomous driving, robotics, etc (Baur et al., 2025; Liang et al., 2024a; Li & Ibanez-Guzman, 2020). These sensors work by periodically emitting laser beams and capturing their reflections,

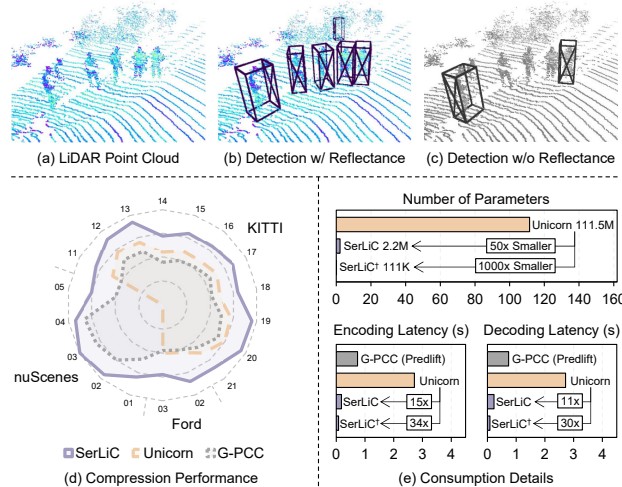

*Figure 1.* (a-c) Reflectance plays an indispensable role in downstream tasks such as 3D object detection; (d) The proposed SerLiC establishes a new state-of-the-art by outperforming the latest compression standard G-PCC (Zhang et al., 2024b) and the learning-based work Unicorn on widely accepted datasets KITTI (Behley et al., 2019), Ford (Pandey et al., 2011), and nuScenes (Caesar et al., 2020) (the number denotes sequence number); (e) SerLiC achieves ultra-low coding latency with an exceptionally lightweight model.

producing vast amounts of data points that form the point cloud—a digital representation of the surrounding 3D environment for downstream perception tasks (Chen et al., 2024; Hu et al., 2023). The massive volume of such data, often reaching gigabytes per minute, highlights the development of efficient LiDAR point cloud compression (PCC) techniques to enable effective storage and transmission (Li et al., 2024; Zhang et al., 2024b; Liu et al., 2020).

Point cloud reflectance, derived from the intensity of returned laser pulses in LiDAR, serves as a crucial descriptor of the physical properties of objects. It is widely used in downstream tasks such as object detection and semantic segmentation (Viswanath et al., 2024; Tatoglu & Pochiraju, 2012). For instance, our experiments show that removing the reflectance causes a dramatic drop in pedestrian (and cyclist) detection accuracy incorporating the pre-trained PointPillar (Lang et al., 2019) detection model, having AP (average precision) from 51.4 (62.8) to 14.1 (34.3) on the KITTI dataset (see Figure 1), making it impractical for use.

[*]Equal contribution [1]School of Information Science and Technology, Hangzhou Normal University, Hangzhou, China [2]School of Electronic Science and Engineering, Nanjing University, Nanjing, China. Correspondence to: Dandan Ding <DandanDing@hznu.edu.cn>.

*Proceedings of the 42nd International Conference on Machine Learning*, Vancouver, Canada. PMLR 267, 2025. Copyright 2025 by the author(s).

While existing neural compression methods have achieved significant performance gains, they mostly focus on the geometry or RGB color attributes (Wang et al., 2025; You et al., 2024; Zhang et al., 2023; Wang et al., 2023), and the compression of LiDAR reflectance is relatively underexplored with uncompetitive efficiency. This compression gap is from the intrinsic challenges of reflectance data.

The first challenge lies in the inherent sparsity of LiDAR scans. Unlike object point clouds comprising relatively dense points, LiDAR point clouds are much sparser, as laser pulses are reflected only from a limited subset of distant points in the environment (Kong et al., 2023; Raj et al., 2020). To address this, convolution-based methods (Wang et al., 2025; 2023) have to employ large-kernel convolutions (e.g., $7^3$) to increase the receptive field to aggregate sufficient points in 3D space for analysis. However, this incurs an expensive computational burden while yielding only marginal performance gains, as the fixed-size large kernel remains insufficient to capture dynamic context.

The second challenge is the intricate physics of underlying LiDAR reflectance. Reflectance intensity is influenced by various factors, such as the material properties of objects, the angle of incidence of laser pulses, the distance between the sensor and the target, and hardware variations (Liang et al., 2024a; Viswanath et al., 2024; Fang et al., 2014). These introduce intertwined correlations across points that extend beyond simple spatial proximity. For instance, reflectance values from surfaces with similar material properties may exhibit strong correlations even when spatially distant. However, existing methods largely neglect these physical interactions inherent in LiDAR reflectance. They typically adopt modeling strategies designed for color attributes in object point clouds and rely on geometric 3D spatial proximity to define LiDAR point relationships. Consequently, their performance remains limited.

Therefore, this work proposes a serialization-based neural compression framework, dubbed SerLiC, that utilizes sequence modeling to improve the efficiency of LiDAR reflectance compression. In contrast to current techniques that directly work in 3D space for spatial correlation exploration, SerLiC follows the LiDAR scan order to serialize 3D points into 1D point sequences, for which expensive 3D operations are removed. The serialization aligns with the LiDAR laser scanning mechanism, allowing for a device-centric rather than spatial perspective in reflectance analysis. Upon this representation, we devise an entropy model to capture point dependencies in a window of each sequence for effective contextual modeling. Within each window, the selective state space model (a.k.a. Mamba (Gu & Dao, 2023)) is employed for autoregressive coding. Additionally, LiDAR scanning-relevant information is derived based on the input point cloud as context to improve the modeling accuracy.

Extensive experiments on benchmark datasets show that SerLiC achieves remarkably high performance, over 2× volume reduction against the original reflectance data and up to 22% bit rate reduction compared to the state-of-the-art Unicorn method (Wang et al., 2025), while using only 2% of the parameters and 10% of the GPU memory, as illustrated in Figure 1. Moreover, SerLiC is hardware-friendly, relying solely on simple networks rather than specific convolution libraries (e.g., Minkowski Engine (Choy et al., 2019)) commonly used in existing methods. Our key contributions are summarized as follows:

- We propose SerLiC, a lossless reflectance compression method for LiDAR point clouds, leveraging scan-order serialization to transform a 3D point cloud to 1D point sequences for efficient representation.

- We generate LiDAR information (scanning index and radial distance) for each point, along with the previous decoded reflectance, as context to exploit point dependencies in a sequence, supported by the selective state space model with a dual parallelization mechanism.

- SerLiC delivers notable performance on benchmark datasets, offering high compression efficiency, ultra-low complexity, and strong robustness. Its light version runs 30 fps with frame pipelining and 10 fps without, with only 111K model parameters.

## 2. Related Work

### 2.1. Serialization-based Point Cloud Analysis

**Transformers.** Serialization has shown remarkable efficacy in point cloud analysis tasks (Wu et al., 2024; Liu et al., 2023; Wang, 2023), owing to the inherent simplicity and computational efficiency of structured data representations. Notable studies, including OctFormer (Wang, 2023), FlatFormer (Liu et al., 2023), and Point Transformer v3 (Wu et al., 2024), have utilized serialization-based Transformer architectures to implement attention mechanisms in structured spaces, achieving high-performance representation. However, Transformers bring quadratic computational complexity, which poses challenges in long sequence modeling (Liu et al., 2024).

**Mamba.** Recently, Mamba (Gu & Dao, 2023) introduced the selective state space model (SSM) mechanism, garnering significant attention for its linear computational complexity and superior performance. Numerous works (Liang et al., 2024b; Zhang et al., 2025; Han et al., 2024; Zhang et al., 2024a; Wang et al., 2024) have successfully integrated Mamba into point cloud analysis to enhance efficiency. For instance, PointMamba (Liang et al., 2024b) applies selective SSM in point cloud analysis by reorganizing key point features into sequences for efficient processing within the

Mamba framework. Extending this paradigm, several studies (Zhang et al., 2024a; Zeng et al., 2024) have introduced Mamba to LiDAR point clouds. However, their serialization approaches are limited to traditional space-filling curves (Morton, 1966; Hilbert, 2013), overlooking unique scanning mechanisms and physical information inherent in LiDAR point clouds.

## 2.2. Point Cloud Attribute Compression

**Rules-based Methods**. Recent advancements in Point Cloud Attribute Compression (PCAC) have led to the standardization of techniques, with MPEG's G-PCC (Zhang et al., 2024b) serving as the benchmark. G-PCC incorporates two primary methods for LiDAR reflectance compression: Region-Adaptive Hierarchical Transform (RAHT) (De Queiroz & Chou, 2016) and Predicting/Lifting Transform (Predlift) (Mammou et al., 2018). To address G-PCC's latency and complexity, L3C2 (Sébastien & Jonathan, 2021) was introduced in 2021, specifically tailored for LiDAR point clouds. However, its dependence on detailed sensor configurations (e.g., the precise pitch angles of individual lasers) limits its applicability.

**Learning-based Methods**. Neural models have revolutionized PCAC by incorporating deep-learning techniques to capture attribute dependencies. CNeT (Nguyen & Kaup, 2023) employs an autoregressive framework with causal priors for sequential prediction, achieving high compression gains at the expense of extremely high computational cost. Building on this, MNeT (Nguyen et al., 2023) adopts a multiscale strategy to enhance computational efficiency but the compression gain is largely compromised. Additionally, CNeT and MNeT are deigned for color attributes, failing to compress the LiDAR reflectance. PoLoPCAC (You et al., 2024) introduces a point-based compression pipeline that models attributes in groups based on antecedents; however, it shows certain dependence on training data, limiting its generalizability. Unicorn (Wang et al., 2025), the latest framework, introduces a universal multiscale coding mechanism using 3D sparse convolution library — Minkowski Engine (Choy et al., 2019) to predict attribute values across multiple scales, achieving state-of-the-art performance. To attain high performance, Unicorn devise large kernel sizes in 3D convolution, resulting in intensive complexity.

**Remarks**. While these learning-based methods have demonstrated promising performance improvements, they mainly focus on general-purpose attributes (e.g., color intensities), which limits their effectiveness in specialized scenarios such as LiDAR reflectance. As the industrial adoption of LiDAR technology grows across various applications like autonomous driving and urban planning (Liang et al., 2024a), developing specialized codecs for LiDAR point clouds is essential to meet practical requirements.

## 3. Method

### 3.1. Problem Definition

Given a LiDAR point cloud with $N$ points, we denote its reflectance intensities as $\mathbf{X} = \{\mathbf{x}_i\}_{i=1}^{N}$ and the geometry coordinates as $\mathbf{C} = \{\mathbf{c}_i\}_{i=1}^{N}$. Here, $\mathbf{x}_i$ and $\mathbf{c}_i$ represent the reflectance attribute and the spatial coordinate of the $i$-th point. Existing approaches (Wang et al., 2025; 2023; You et al., 2024) rely exclusively on the *unordered* geometric coordinates $\mathbf{C}$ to model point correlations, thereby neglecting the inherent sequential scanning characteristics of LiDAR data. In contrast, this paper introduces **serialization** for leveraging the sequential nature of LiDAR scans. This is because 1) Serialization transforms point clouds into a 1D representation, enabling more efficient and structured modeling from a device-centric perspective; and 2) Serialization supports the efficient processing of information that more closely reflects the physical characteristics of LiDAR scanning ($\mathbf{C} \rightarrow \mathbf{C}^*$), resulting in more accurate analysis.

The serialization process reorganizes the input point cloud $(\mathbf{X}, \mathbf{C})$ into $L$ point sequences as follows (see Figure 2):

$$\{(\mathbf{X}_l, \mathbf{C}_l^*)\}_{l=1}^{L} = \text{Serialize}\,(\mathbf{X}, \mathbf{C})\,, \qquad (1)$$

where $L$ denotes the number of laser beams emitted by the LiDAR sensor. Each sequence $\mathbf{X}_l = \{\mathbf{x}_{l,i}\}_{i=1}^{N_l}$ represents the reflectance intensities from the $l$-th laser beam, and $\mathbf{C}_l^* = \left\{\mathbf{c}_{l,i}^*\right\}_{i=1}^{N_l}$ encodes the contextual information of the corresponding points.

In this context, we define LiDAR information $\mathbf{c}_{l,i}^* = (v_{l,i}, u_{l,i}, \rho_{l,i})$, including the laser index $v_{l,i}$, the azimuth angle index $u_{l,i}$, and the radial distance $\rho_{l,i}$ for each point. These information can be entirely derived from the geometric coordinates $\mathbf{C}$ during the serialization process, as detailed in Section 3.2.

With the serialized representation $(\mathbf{X}_l, \mathbf{C}_l^*)$, our objective is to design a parameterized entropy model that incorporates the priors $\mathbf{C}_l^*$ to exploit spatial and contextual dependencies in the reflectance data $\mathbf{X}_l$. This model aims to approximate the conditional probability distribution $p(\mathbf{X}_l|\mathbf{C}_l^*)$ as closely as possible to the true distribution $q(\mathbf{X}_l|\mathbf{C}_l^*)$. The total encoded bit rate $R$ is the sum of bit rates for all sequences, which can be expressed using the Shannon cross-entropy between two distributions:

$$R = \sum_{l=1}^{L} R_l = \sum_{l=1}^{L} \mathbb{E}_{\mathbf{X}_l \sim q(\mathbf{X}_l|\mathbf{C}_l^*)}[-\log_2 p(\mathbf{X}_l|\mathbf{C}_l^*)]. \quad (2)$$

### 3.2. Serialization

Serialization has demonstrated significant efficacy in a wide range of point cloud analysis tasks, owing to the inherent simplicity and computational efficiency of structured

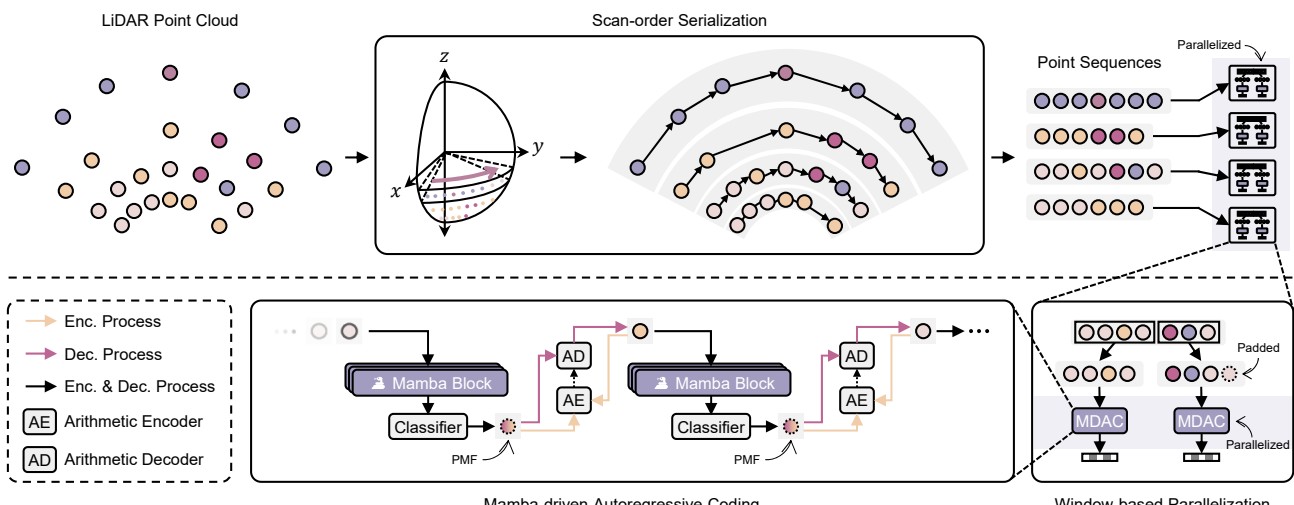

*Figure 2.* **SerLiC Framework.** The input 3D LiDAR point cloud is first serialized into 1D ordered point sequences, which are then divided into windows for parallel processing. For each window, a Mamba-driven autoregressive coding (MDAC) scheme is employed, which embeds scanning index ($\mathbf{F}_i^{pos}$), radial distance ($\mathbf{F}_i^{\rho}$), and prior reflectance ($\mathbf{F}_{i-1}^{\mathbf{x}}$) as context to generate the probability mass function (PMF) for the reflectance intensity of the target ($i$-th) point.

data (Liang et al., 2024b; Zhang et al., 2025; Wu et al., 2024). Unlike conventional point cloud serialization strategies, such as the space-filling curves (Morton, 1966; Hilbert, 2013; Liang et al., 2024b; Zhang et al., 2025), our proposed SerLiC introduces a LiDAR scan-order serialization, specifically designed to effectively preserve and leverage the intrinsic regularities within a LiDAR point cloud.

**Coordinate Mapping.** Specifically, the Cartesian coordinate $\mathbf{c}_i$ of a point is first converted to spherical coordinate space, yielding the elevation angle $\varphi_i \in \left[-\frac{\pi}{2}, \frac{\pi}{2}\right]$, azimuth angle $\phi_i \in (-\pi, \pi]$, and radial distance $\rho_i \in (0, +\infty)$[1]. The transformation is expressed as:

$$\begin{cases} \rho_i = \sqrt{x_i^2 + y_i^2 + z_i^2} \\ \varphi_i = \arcsin\left(\dfrac{z_i}{\rho_i}\right) \\ \phi_i = \operatorname{atan2}(y_i, x_i) \end{cases}, \quad (3)$$

where $x_i$, $y_i$, and $z_i$ denote the Cartesian coordinate of $\mathbf{c}_i$. Next, each point is mapped to a discrete grid by assigning a laser index $v_i$ and an azimuthal index $u_i$, respectively, based on its elevation angle $\varphi_i$ and azimuth angle $\phi_i$:

$$\begin{aligned} v_i &= \left\lfloor L \times \left(\frac{\varphi_i - \varphi_{\text{down}}}{\varphi_{\text{up}} - \varphi_{\text{down}}}\right)\right\rfloor + 1, \\ u_i &= \left\lfloor W \times \left(\frac{\phi_i}{2\pi} + \frac{1}{2}\right)\right\rfloor + 1, \end{aligned} \quad (4)$$

---

[1]In practical systems, the maximum detection range of LiDAR is constrained by sensor specifications, typically up to 400 meters (Liang et al., 2024a).

where $L$ and $W$ represent the angular resolutions of the elevation and azimuth angles, respectively; $L$ is the number of laser channels; $\lfloor \rfloor$ denotes the floor operation; $\varphi_{\text{up}}$ and $\varphi_{\text{down}}$ are the maximum and minimum elevation angles, defining the upward and downward view field of LiDAR sensor. Notably, the values of $v_i$ and $u_i$ are clipped to ranges $[1, L]$ and $[1, W]$, respectively. As such, the contextual information $\mathbf{c}_i^*$ of each point is aggregated as $(v_i, u_i, \rho_i)$.

**Reordering.** Based on the angular indices $v_i$ and $u_i$, the point cloud data are organized and sorted as follows:

- Points with the same laser index $v_i = l(l \in [1, L])$ are aggregated into subsets $(\tilde{\mathbf{X}}_l, \tilde{\mathbf{C}}_l^*)$, with $(\tilde{\mathbf{X}}_l, \tilde{\mathbf{C}}_l^*) = \{(\mathbf{x}_i, \mathbf{c}_i^*) \mid v_i = l, i \in [1, N]\}$.

- Within each subset, points are ordered by increasing azimuthal index $u_i$, yielding a sequence $(\mathbf{X}_l, \mathbf{C}_l^*) = \{(\mathbf{x}_i, \mathbf{c}_i^*) \in (\tilde{\mathbf{X}}_l, \tilde{\mathbf{C}}_l^*) \mid u_i \le u_j, \forall i < j\}$.

In this way, the proposed serialization method restructures the point cloud into internally ordered sequences, enabling efficient autoregressive coding using the state-space model.

### 3.3. Contextual Construction

Figure 2 illustrates the state space model-based autoregressive encoder for sequentially compressing the reflectance attributes of the point sequence. Each point is tokenized into a contextual combination of scanning index, radial distance, and prior reflectance, with the aim of exploring correlations to its neighbor for accurate probability estimation.

**Scanning Index.** Previous methods usually explicitly leverage geometry coordinates to identify correlated neighbors for a point. However, spatially close points may not be highly correlated as the LiDAR points are derived following its unique scanning mechanism. To this end, leveraging the scanning index for point context will yield better results since points with the same index are captured similarly. Specifically, the scanning index feature $\mathbf{F}_i^{pos}$ of the $i$-th point is derived by directly embedding the laser and azimuth indices (i.e., $v_i$ and $u_i$) obtained in Section 3.2:

$$\mathbf{F}_i^{pos} = \text{Embed}\,(v_i) \oplus \text{Embed}\,(u_i)\,,\; 1 \le i \le N_l, \quad (5)$$

where $\text{Embed}$ denotes the embedding layer, and $\oplus$ refers to the concatenate operation.

**Radial Distance.** LiDAR reflectance essentially represents the backscattered intensity of LiDAR signals, which are intrinsically influenced by the distance between the object and the sensor, based on the light transmission theory (Viswanath et al., 2024; Fang et al., 2014). Therefore, SerLiC explicitly incorporates radial distance as context to enhance the reflectance compression. Specifically, the radial distance $\rho_i$ (see Eq. (3)) is normalized and transformed into a high-dimensional spatial feature $\mathbf{F}_i^\rho$ using a Linear layer:

$$\mathbf{F}_i^\rho = \text{Linear}\,(\text{Norm}\,(\rho_i))\,,\; 1 \le i \le N_l, \quad (6)$$

where $\text{Norm}$ refers to the min-max normalization that rescales $\rho_i$ to the range of $(0,1)$.

**Prior Reflectance.** Naturally, the reflectance value of previous point in the sequence is embedded ($\mathbf{F}_{i-1}^{\mathbf{x}}$) as context.

We then combine all context for subsequent processing:

$$\mathbf{F}_i^{token} = \left(\mathbf{F}_i^{pos} \oplus \mathbf{F}_i^\rho \oplus \mathbf{F}_{i-1}^{\mathbf{x}}\right),\; 1 \le i \le N_l. \quad (7)$$

For the case of $i{=}1$, where prior reflectance is unavailable, the value of $\mathbf{F}_0^{\mathbf{x}}$ is estimated by the neural network.

### 3.4. Entropy Coding

**Mamba Coder.** Given the collected context token $\mathbf{F}_i^{token}$ of a point sequence, Mamba is ideally suited for autoregressive coding due to its ability to model sequential dependencies and capture correlations between points.

Thus, the token $\mathbf{F}_i^{token}$ is passed through several Mamba blocks. A standard Mamba layer is shown in Figure 3(a), and the processing flow can be summarized as follows:

$$
\begin{aligned}
\bar{\mathbf{F}}_i^s &= \text{LayerNorm}\left(\mathbf{F}_i^{s-1}\right), \\
\tilde{\mathbf{F}}_i^s &= \sigma\left(\text{DWConv}\left(\text{Linear}\left(\bar{\mathbf{F}}_i^s\right)\right)\right), \\
\hat{\mathbf{F}}_i^s &= \sigma\left(\text{Linear}\left(\bar{\mathbf{F}}_i^s\right)\right), \\
\mathbf{F}_i^s &= \text{Linear}\left(\text{SelectiveSSM}(\tilde{\mathbf{F}}_i^s) \otimes \hat{\mathbf{F}}_i^s\right) + \mathbf{F}_i^{s-1},
\end{aligned}
\quad (8)
$$

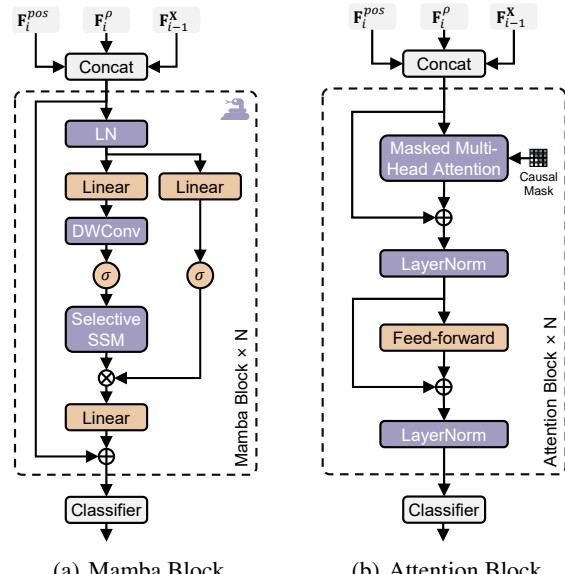

(a) Mamba Block      (b) Attention Block

*Figure 3.* Basic Mamba block and Attention block. "LN" refers to Layer Norm; $\sigma$ denotes SiLU activation; $\otimes$ means Hadamard product; $\oplus$ represents element-wise addition.

where $\mathbf{F}_i^s$ denotes the output of the $s$-th Mamba layer; DWConv represents the depth-wise convolution; $\sigma$ refers to the SiLU activation function; $\otimes$ means Hadamard product. The initial input $\mathbf{F}_i^0$ is set to the token feature $\mathbf{F}_i^{token}$.

Let $\mathbf{F}_i^{out}$ be the output of the final Mamba layer. Then a classifier is used to predict the probability mass function (PMF) for the target ($i$-th) point, leveraging the Linear transform and Softmax function:

$$p_{l,i}\left(\cdot \mid \mathbf{x}_{l,<i}, \mathbf{c}_{l,\le i}^*\right) = \text{Softmax}\left(\text{Linear}\left(\mathbf{F}_i^{out}\right)\right). \quad (9)$$

**Dual Parallelization.** SerLiC proposes a dual parallelization strategy at both the sequence and window levels to accelerate the coding process. First, as shown in Figure 2, SerLiC processes point sequences in parallel, disregarding correlations across sequences. This is justified, as point sequences are scanned by separate laser sensor rotations, inherently lacking strong inter-sequence dependencies. As a result, sequence parallelization does not compromise coding efficiency but greatly enhances processing speed.

Furthermore, SerLiC introduces window-based parallelization within each point sequence. LiDAR scanning systems, particularly those utilizing mechanical rotation, produce sequences with extensive spatial coverage, often spanning a full 360-degree field of view. Within such sequences, spatial regions exhibit varying degrees of correlation—local regions like the area directly in front of the sensor have stronger internal correlations than distant regions located behind or to the sides. This localized correlation characteristic enables

Table 1. Quantitative compression gains against other methods

| CLASS | SEQ. | BITS PER POINT (BPP) | | | | | COMPRESSION GAIN | | | |
| | | G-PCC (RAHT) | G-PCC (PREDLIFT) | UNICORN | SERLIC (LIGHT) | SERLIC | VS. RAHT | VS. PREDLIFT | VS. UNICORN | VS. LIGHT |
|---|---|---|---|---|---|---|---|---|---|---|
| KITTI | 11 | 5.10 | 5.01 | 4.59 | 4.06 | 3.80 | -25.49% | -24.15% | -17.21% | -6.40% |
| | 12 | 4.56 | 4.51 | 4.14 | 3.68 | 3.45 | -24.34% | -23.50% | -16.67% | -6.25% |
| | 13 | 4.79 | 4.62 | 4.15 | 3.63 | 3.35 | -30.06% | -27.49% | -19.28% | -7.71% |
| | 14 | 5.24 | 5.22 | 4.82 | 4.35 | 4.16 | -20.61% | -20.31% | -13.69% | -4.37% |
| | 15 | 5.11 | 5.03 | 4.61 | 4.13 | 3.87 | -24.27% | -23.06% | -16.05% | -6.30% |
| | 16 | 5.00 | 4.96 | 4.55 | 4.11 | 3.85 | -23.00% | -22.38% | -15.38% | -6.33% |
| | 17 | 4.80 | 4.73 | 4.38 | 3.87 | 3.64 | -24.17% | -23.04% | -16.89% | -5.94% |
| | 18 | 5.06 | 4.97 | 4.56 | 4.08 | 3.80 | -24.90% | -23.54% | -16.67% | -6.86% |
| | 19 | 4.63 | 4.53 | 4.11 | 3.50 | 3.20 | -30.89% | -29.36% | -22.14% | -8.57% |
| | 20 | 4.51 | 4.35 | 3.98 | 3.52 | 3.28 | -27.27% | -24.60% | -17.59% | -6.82% |
| | 21 | 4.83 | 4.78 | 4.41 | 3.93 | 3.69 | -23.60% | -22.80% | -16.33% | -6.11% |
| | **AVG.** | **4.88** | **4.79** | **4.39** | **3.90** | **3.64** | **-25.41%** | **-24.01%** | **-17.08%** | **-6.67%** |
| FORD | 02 | 5.18 | 5.03 | 4.89 | 4.29 | 3.57 | -31.08% | -28.94% | -26.99% | -16.78% |
| | 03 | 5.13 | 5.05 | 5.04 | 4.46 | 4.11 | -19.91% | -18.65% | -18.45% | -7.85% |
| | **AVG.** | **5.16** | **5.04** | **4.97** | **4.29** | **3.84** | **-25.58%** | **-23.81%** | **-22.74%** | **-12.33%** |
| NUSCENES | 01 | 3.93 | 3.61 | - | 3.13 | 2.89 | -26.46% | -19.94% | - | -4.30% |
| | 02 | 3.41 | 3.08 | - | 2.46 | 2.58 | -33.14% | -25.97% | - | -7.32% |
| | 03 | 3.18 | 2.77 | - | 2.23 | 2.34 | -34.59% | -24.91% | - | -6.73% |
| | 04 | 3.05 | 2.69 | - | 2.28 | 2.41 | -22.62% | -12.27% | - | -3.51% |
| | 05 | 4.40 | 4.04 | - | 3.23 | 3.33 | -31.82% | -25.74% | - | -7.12% |
| | **AVG.** | **3.59** | **3.24** | **-** | **2.78** | **2.52** | **-29.81%** | **-22.22%** | **-** | **-9.35%** |

Table 2. Quantitative compression gains against L3C2

| CLASS | SEQ. | BPP | | CR GAIN |
| | | L3C2 | SERLIC | VS. L3C2 |
|---|---|---|---|---|
| FORD | 02 | 4.83 | 3.57 | -26.15% |
| | 03 | 4.98 | 4.11 | -17.39% |
| | **AVG.** | **4.90** | **3.84** | **-21.63%** |

effective compression even when processing is limited to smaller windows rather than the entire sequence. Accordingly, SerLiC partitions each sequence into independent windows for parallel processing, ensuring both efficiency and performance. Figure 2 depicts the window slicing method. Each sequence is partitioned equally. Padding points are appended at the end to maintain a uniform window size for consistent processing.

## 4. Experiment and Analysis

### 4.1. Datasets

We conducted experiments on well-known LiDAR datasets, including KITTI (Behley et al., 2019), Ford (Pandey et al., 2011), and nuScenes (Caesar et al., 2020).

- **KITTI** or SemanticKITTI is a large-scale LiDAR dataset used for semantic scene understanding. It contains 22 sequences, a total of 43,552 frames of outdoor scenes collected using the Velodyne HDL-64E LiDAR

sensor. There are around 120k points on average per frame. These raw floating-point coordinates are quantized to 1mm precision (18 bits) with 7-bit reflectance.

- **Ford** is also collected using Velodyne HDL-64E. The common test condition (CTC) defined by MPEG (WG 07 MPEG 3D Graphics Coding and Haptics Coding, 2024b) utilizes three Ford sequences, each having 1,500 frames at 1mm precision with 8-bit reflectance. The first sequence is for training, and the remaining two are for testing.

- **nuScenes** is a large-scale dataset collected for autonomous driving using Velodyne HDL-32E. It has ten subsets, each containing 85 scenes. For training, we extract the first 100 frames from the first 12 scenes in the first five subsets, resulting in 6,000 frames. For testing, we select the first 90 frames from the first scene of each of the last five subsets, yielding 450 frames numbered as sequences #01 to #05. Also, we quantize them to 1mm geometric precision with 8-bit reflectance.

### 4.2. Experimental Details

**Training Setting.** We implement SerLiC using Python 3.10 and PyTorch 2.5. The model is trained with AdamW optimizer (Loshchilov & Hutter, 2019), using a learning rate of $2 \times 10^{-4}$ and a batch size of 64. We employ a cosine annealing strategy (Loshchilov & Hutter, 2017) to gradually reduce the learning rate to $5 \times 10^{-5}$. The model is randomly initialized and trained for 25 epochs for each dataset. For

fair comparisons, all experiments are conducted on the same platform, equipped with an NVIDIA RTX 4090 GPU, an Intel Core i9-13900K CPU, and 64GB of memory.

The optimization objective is to minimize the bit rate (as defined in Equation (2)) for transmitting reflectance data, which can be further formulated as the negative log-likelihood of the observed reflectance values across all sequences $l \in [1, L]$ and points $i \in [1, N_l]$:

$$\mathcal{L} = -\sum_{l=1}^{L} \sum_{i=1}^{N_l} \log_2 p_{l,i}\left(x_{l,i} \mid \text{context}_{l,i}\right), \quad (10)$$

where $p_{l,i}\left(x_{l,i} \mid \text{context}_{l,i}\right)$ denotes the conditional probability estimated by the parametrized entropy model $p_{l,i}$ based on the $\text{context}_{l,i}$.

**Testing Setting.** The testing conditions strictly follow the CTC of MPEG AI-PCC (WG 07 MPEG 3D Graphics Coding and Haptics Coding, 2024a). Quantitative analysis is measured in bits per point (bpp) and compression ratio (CR). The latest G-PCC version TMC13v23[2], which provides state-of-the-art performance through RAHT and Predlift modes, is compared. Also, we compare SerLiC with L3C2 (Sébastien & Jonathan, 2021), a profile dedicated for LiDAR PCC in MPEG, and existing learning-based Unicorn (Wang et al., 2025) which shows superior performance on LiDAR reflectance compression. All methods are evaluated under the same training/testing datasets and conditions.

### 4.3. Compression Performance

Table 1 presents a detailed comparison of the overall bit rate and CR gains of SerLiC against G-PCC (RAHT), G-PCC (Predlift), and Unicorn. As observed, SerLiC consistently outperforms both G-PCC (RAHT) and G-PCC (Predlift), across all three tested datasets, e.g., SerLiC rivals G-PCC (RAHT) by 25.41%, 25.58%, and 29.81% on three datasets. Substantial performance improvements (17.08% on KITTI and 22.74% on Ford) are observed when compared with Unicorn and L3C2 (21.63% on Ford).

Notice that the compression of L3C2 relies heavily on detailed parameters of LiDAR sensor, including the elevation angle of each laser sensor and their respective distances to the LiDAR center. For datasets lacking such sensor data, such as KITTI and nuScenes, L3C2 cannot function effectively (hence we only provide L3C2 results on Ford). In contrast, SerLiC only requires basic LiDAR information such as the angular resolutions of the elevation and azimuth angles. These results are evidence of the superior effectiveness of SerLiC as well as its robustness on various datasets.

[2] https://github.com/MPEGGroup/mpeg-pcc-tmc13

*Table 3.* Computational complexity on the KITTI dataset

| METHOD | MEM. | PARAM. | ENC. | DEC. |
|---|---|---|---|---|
| RAHT | - | - | 0.36s | 0.34s |
| PREDLIFT | - | - | 0.74s | 0.74s |
| UNICORN | 4.36GB | 111.5M | 2.77s | 2.77s |
| SERLIC | 0.41GB | 2.2M | 0.09s*/0.18s | 0.13s*/0.23s |
| SERLIC (LIGHT) | 0.25GB | 111K | 0.03s*/0.08s | 0.03s*/0.09s |

\* RESULTS OF USING THE FRAME-LEVEL PIPELINING

### 4.4. Computational Complexity

Table 3 reports the computation complexity. SerLiC uses only 2.2M parameters, which is only 2% of Unicorn.

For runtime, the coding process consists of three components: context construction (CC, on CPU), neural network (NN, on GPU), and arithmetic coding (AC, on CPU). By utilizing a three-stage pipeline in frame-level to arrange these components, SerLiC achieves an encoding/decoding speed of 0.09/0.13 seconds per frame, i.e., >11/7 frames per second (fps). Even when the frame-level pipelining is disabled, the total runtime remains much shorter (<10%) than that of Unicorn, achieving 4 to 5 fps. Note that our experiments use point clouds with 18-bit geometry, containing hundreds of thousands of points per frame. If applied to 12-bit point clouds, which have far fewer points but still maintain high accuracy for downstream tasks, the coding speed of SerLiC would be even faster. *Detailed runtime and analysis can be found in our supplementary material.*

Regarding GPU memory usage, SerLiC uses 0.41 GB while Unicorn uses 4.36 GB, only 9% of Unicorn. The number of parameters in SerLiC is also much smaller: the full version is only 2% of Unicorn while the light version is only 1‰. All these confirm the ultra-low complexity of SerLiC.

### 4.5. Ablation Study

Ablation studies are conducted on the KITTI dataset to validate SerLiC. Default settings are marked in  gray . The results of CR gains are over the G-PCC (RAHT) anchor.

*Table 4.* Ablation study on contextual construction

| VARIANTS | | | BPP | CR GAIN |
|---|---|---|---|---|
| $+ \mathbf{F}_{i-1}^{\mathbf{x}}$ | $+ \mathbf{F}_i^{\rho}$ | $+ \mathbf{F}_i^{pos}$ | | |
| ✓ | ✗ | ✗ | 3.92 | -19.67% |
| ✓ | ✗ | ✓ | 3.82 | -21.72% |
| ✓ | ✓ | ✗ | 3.73 | -23.57% |
| ✓ | ✓ | ✓ | **3.64** | **-25.41%** |

**Contextual Construction**. When certain components are disabled, we expand the embedding dimension of the remaining components to ensure that the overall network di-

mension entering the Mamba block remains consistent. In Table 4, when both radial distance and scanning index embeddings are disabled, the only available prior information is the reflectance of previous point $\mathbf{F}^{\mathbf{x}}_{i-1}$. As a result, the gains decrease from 25.41% to 19.67%. Also, disabling either $\mathbf{F}^{\rho}_i$ or $\mathbf{F}^{pos}_i$ causes loss. These results confirm the significance of utilizing LiDAR physical information.

*Table 5.* Ablation study on parallel window size, maximum GPU memory (Mem.), neural network (NN) latency (encoding/decoding), and total coding time (encoding/decoding)

| SIZE | BPP | MEM. (GB) | NN (S) | TOTAL (S) |
|------|-----|-----------|--------|-----------|
| 64 | 3.68 | 0.61 | **0.08/0.09** | **0.16/0.18** |
| 128 | **3.64** | 0.41 | 0.09/0.13 | 0.18/0.23 |
| 256 | **3.64** | 0.32 | 0.17/0.28 | 0.32/0.43 |
| 512 | **3.64** | 0.29 | 0.33/0.57 | 0.57/0.60 |
| 1024 | 3.66 | **0.28** | 0.66/1.03 | 1.07/1.43 |

**Window-based Parallelization**. We further investigate the impact of window-based parallelization on encoding/decoding runtime and memory usage. For fair comparisons, we disable the frame pipelining in this ablation. Table 5 presents the results for various parallel window sizes, ranging from 64 to 1024. Increasing the window size substantially extends the encoding and decoding time due to the involvement of more autoregressive steps in a window. At the same time, GPU memory usage decreases as the number of parallel windows is reduced. As a result, we set the window size as 128 by default for SerLiC, striking a balance between coding time and memory consumption.

*Table 6.* Ablation study on the Mamba network

| MODULE | SETS. | PARAM. | BPP | CR GAIN |
|--------|-------|--------|-----|---------|
| LAYERS | 1 | 488K | 3.84 | -21.31% |
| | 3 | 1.4M | 3.68 | -24.48% |
| | 5 | 2.2M | 3.64 | -25.41% |
| | 7 | 3.1M | 3.61 | -26.02% |
| DIMENSION | 64 | 176K | 3.76 | -22.95% |
| | 128 | 609K | 3.69 | -24.39% |
| | 256 | 2.2M | 3.64 | -25.41% |
| | 512 | 8.6M | 3.61 | -26.02% |

**Mamba Network**. To study the effect of the number of Mamba layers on model performance, we conducted experiments with layer configurations $\{1, 3, 5, 7\}$. For each configuration, the network dimension is set to 256. As shown in Table 6, the CR gains against G-PCC (RAHT) increase from 21.31% to 26.02% as the number of layers increases from 1 to 7. We also investigate the impact of network dimension by setting it to $\{64, 128, 256, 512\}$. Here, the number of Mamba layers is fixed at 5. As reported in Table 6, the gains increase from 22.95% to 26.02% as the network dimension

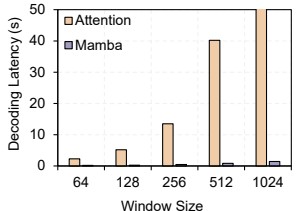 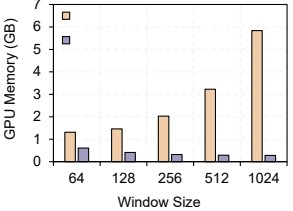

(a) Decoding Latency  (b) Maximum GPU Memory

*Figure 4.* Comparison of decoding latency and running memory between Attention and Mamba implementations in SerLiC.

increases from 64 to 512. These results are in line with the general expectation that larger models have better capacity. However, higher complexity is required.

**Attention-based Coding.** Attention represents another prevalent neural structure for sequence modeling, as exemplified by OctAttention (Fu et al., 2022) and Point Transformer v3 (Wu et al., 2024). For a fair comparison, we substitute the original Mamba module with Masked Multi-Head Attention (Vaswani et al., 2017) (detailed in Figure 3(b)) while maintaining strict consistency in key hyperparameters, such as the number of layers and channels. We observe that using Attention receives similar coding gains to Mamba but higher complexity. Figure 4 illustrates the complexity of Attention-based and Mamba-based SerLiC under different configurations. It is observed that Attention has much higher complexity, in both decoding delay and GPU memory consumption. This is because Attention computes point dependencies within a sequence window $W$ via matrix multiplication, resulting in quadratic complexity $\Theta(W^2)$. Particularly for decoding which requires point-by-point processing in a window, the complexity increases to $\Theta(W^3)$.

In contrast, Mamba consistently maintains linear computational complexity. For example, when the parallel window size is 128, Attention needs 5.12 seconds and 1.46 GB memory for decoding a KITTI point cloud, while Mamba needs only 0.23 seconds with 0.41 GB memory, which is much more beneficial for resource-constrained applications.

**Light Version.** A light version of SerLiC is implemented by reducing the network dimension of standard SerLiC from 256 to 64, the number of layers from 5 to 3, and window size from 128 to 32. As presented in Table 1, our SerLiC (light) receives 6.67% loss against the standard version on the KITTI dataset. However, the computational complexity is greatly reduced, with the number of parameters substantially decreased to just 111K — over 1,000 × smaller than that of Unicorn and the coding speed at > 10 fps (0.08-0.09 seconds per frame) even without the frame pipelining implementation. Under the frame-level pipeline, SerLiC (light) achieves a real-time processing speed of over 30 fps (0.03 seconds per frame).

Table 7. Quantitative compression gains against G-PCC (RAHT) on the non-rotational dataset

| CLASS | SEQ. | BPP | | CR GAIN |
| | | RAHT | SERLIC | VS. RAHT |
|---|---|---|---|---|
| INNOVIZQC | 02 | 3.71 | 2.28 | -38.54% |
| | 03 | 3.82 | 2.25 | -41.10% |
| | **AVG.** | **3.77** | **2.27** | **-39.71%** |

Table 8. Quantitative compression gains against G-PCC (RAHT) on various density scenarios in KITTI

| CLASS | SEQ. | BPP | | CR GAIN |
| | | RAHT | SERLIC | VS. RAHT |
|---|---|---|---|---|
| KITTI | CITY | 4.67 | 3.24 | -30.62% |
| | RESIDENTIAL | 4.96 | 3.81 | -23.19% |
| | ROAD | 5.41 | 4.39 | -18.85% |
| | CAMPUS | 4.86 | 3.43 | -29.42% |
| | **AVG.** | **4.98** | **3.72** | **-25.28%** |

**Non-Rotational Adaptation.** In addition to the KITTI, Ford, and nuScenes datasets that are captured by rotational LiDAR scanning, we also evaluated SerLiC on the non-rotational InnovizQC dataset provided by MPEG. It provides three sequences, each having 300 frames with 16-bit coordinates and 8-bit reflectance. We used the first sequence for training and the remaining two for testing. Results shown in Table 7 demonstrate SerLiC's strong generalization capability to non-rotational LiDAR systems.

**Scenario Robustness.** To further validate the robustness of SerLiC, we provide its detailed performance on KITTI across various scenarios, as reported in Table 8. Results indicate that SerLiC consistently outperforms G-PCC in all scenes. Relatively, SerLiC achieves better performance in City and Campus scenes compared to Residential and Road scenes. This occurs due to dense roadside vegetation interfering with sensor-recorded reflectance characteristics and thus increasing compression difficulty.

## 5. Conclusion

This paper presents SerLiC, a serialization-based neural compression framework tailored for LiDAR reflectance attribute. By leveraging scan-order serialization, SerLiC transforms 3D point clouds into 1D sequences, aligning with LiDAR scanning mechanisms and enabling efficient sequential modeling. The Mamba model with physics-informed tokenization further enhances its ability to capture points correlations autoregressively, while maintaining linear-time complexity. Its high efficiency, ultra-low complexity, and strong robustness make it a practical solution for real LiDAR applications. Future work will extend SerLiC to lossy compression for higher compression efficiency.

## Impact Statement

This paper presents work whose goal is to advance the field of Machine Learning. There are many potential societal consequences of our work, none which we feel must be specifically highlighted here.

## Acknowledgments

This research was supported by National Natural Science Foundation of China (62171174) and Natural Science Foundation of Jiangsu Province (BK20243038). We are grateful to the anonymous reviewers for comments on early drafts of this paper. We also extend our sincere thanks to the authors of the relevant works used in our comparative studies for providing the latest results for evaluation.

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

## A. Testing Dataset Details

Three widely accepted datasets, including KITTI, Ford, and nuScenes, are used as testing datasets in this work. We visualize typical samples from these three datasets in Figure 1. Accordingly, Table 1 provides their detailed information.

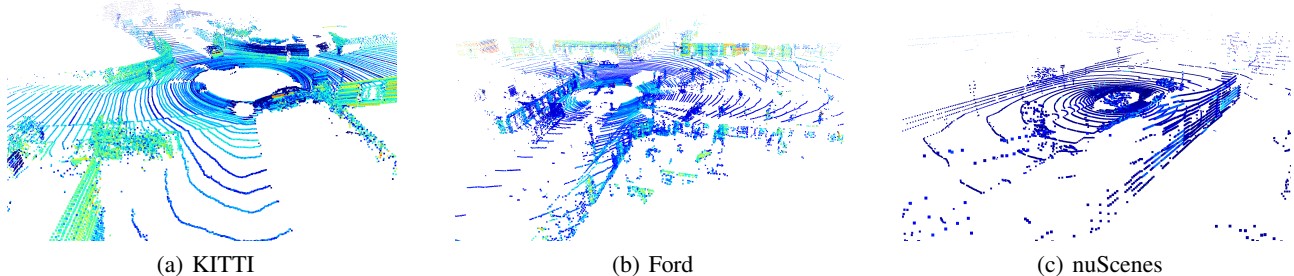

(a) KITTI            (b) Ford            (c) nuScenes

*Figure 1.* Visualization of samples in KITTI, Ford, and nuScenes LiDAR point cloud datasets. The color indicates the value of reflectance, ranging from blue (low) to red (high).

*Table 1.* Detailed information of testing datasets

| CLASS | SEQ. | FRAMES | REFLECTANCE VALUE | POINTS PER FRAME |
|---|---|---|---|---|
| | 11 | 90 | 0-99 | 124,130 |
| | 12 | 90 | 0-99 | 110,921 |
| | 13 | 90 | 0-99 | 111,997 |
| | 14 | 90 | 0-99 | 124,628 |
| | 15 | 90 | 0-99 | 122,315 |
| KITTI | 16 | 90 | 0-99 | 125,622 |
| | 17 | 90 | 0-99 | 113,687 |
| | 18 | 90 | 0-99 | 124,873 |
| | 19 | 90 | 0-99 | 121,239 |
| | 20 | 90 | 0-99 | 116,382 |
| | 21 | 90 | 0-99 | 123,307 |
| FORD | 02 | 1500 | 0-255 | 83,834 |
| | 03 | 1500 | 0-255 | 84,063 |
| | 01 | 90 | 0-255 | 29,606 |
| | 02 | 90 | 0-255 | 29,568 |
| NUSCENES | 03 | 90 | 0-255 | 27,560 |
| | 04 | 90 | 0-255 | 26,854 |
| | 05 | 90 | 0-255 | 30,478 |

## B. Complexity and Analysis

All the computational complexity reported in the following are derived on our experiment platform equipped with an NVIDIA RTX 4090 GPU, an Intel Core i9-13900K CPU, and 64GB of memory.

### B.1. SerLiC Runtime Analysis

The coding process of SerLiC consists of three stages: context construction (CC, executed on CPU), neural network (NN, executed on GPU), and arithmetic coding (AC, executed on CPU). These three stages work following the proposed Dual Parallelization mechanism when processing a frame, including the sequence level and the window level, as illustrated in Figure 2(a). Each sequence is divided into windows of equal size and all windows are processed in parallel.

Based on the above Dual Parallelization within a frame, we offer two ways to implement SerLiC across point cloud frames:

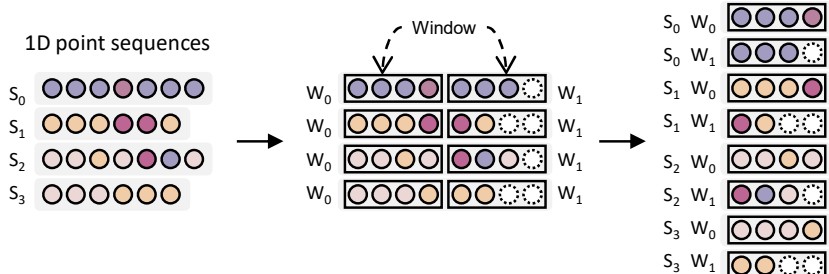

(a) Details of Dual Parallelization in a Frame

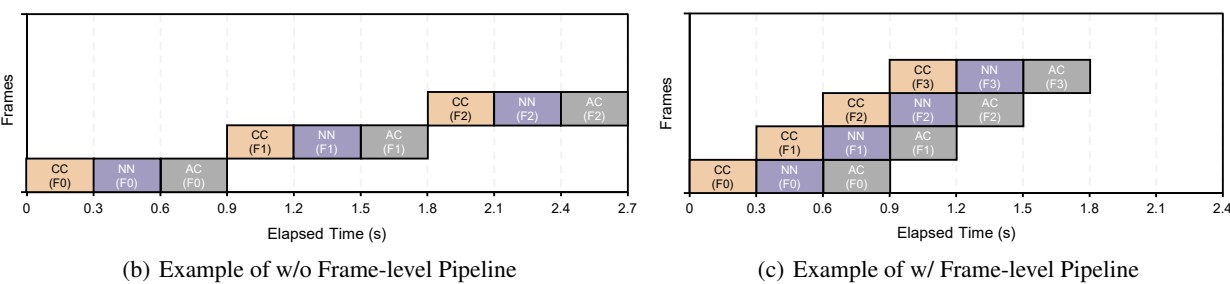

(b) Example of w/o Frame-level Pipeline

(c) Example of w/ Frame-level Pipeline

*Figure 2.* (a) Proposed Dual Parallelization within a frame. (b) Sequential implementation across frames. (c) Frame-level pipeline across frames. $S_i$ indicates the point sequence. $W_i$ denotes the window. $F_i$ denote the $i$-th point cloud frame. We use SerLiC (light) to show the examples in (b) and (c).

- **Frame-level sequential implementation**. We implemented a frame-level sequential version of SerLiC, running point cloud frames sequentially on our platform, without any pipeline structure, as illustrated in Figure 2(b). As reported in Table 2 and Figure 3(a)-(b), our SerLiC attains 0.18/0.23 seconds per frame for encoding/decoding, i.e., >5/4 fps (frames per second). SerLiC (light) achieves a faster speed of >10 fps. Even using the sequential version, our SerLiC achieves state-of-the-art processing speed. Its light version can meet typical demands in applications such as conventional autonomous driving, LiDAR-based mapping, and geospatial surveying.

- **Frame-level pipeline implementation**. To accelerate processing, we arrange CC, NN, and AC on three pipeline stages for frame-level parallelism, as illustrated in Figure 2(c). As presented in Table 2 and Figure 3(c)-(d), using the three-stage pipelining across frames, SerLiC runs approximately at a speed of 0.09/0.13 seconds per frame for encoding/decoding, i.e., 11/7.7 fps. The lightweight version of SerLiC, SerLiC (light), is even faster, achieving over 30 fps, meeting the real-time requirements of many LiDAR-related applications like real-time obstacle detection in autonomous driving, virtual and augmented reality (VR/AR), and high-fidelity 3D reconstruction.

*Table 2.* Analysis of encoding/decoding time (seconds per frame), including context construction (CC) latency, neural network (NN) latency, and arithmetic coding (AC) latency

| METHOD | CC (S) | NN (S) | AC (S) | TOTAL (S) W/ PIPELINE | TOTAL (S) W/O PIPELINE |
|---|---|---|---|---|---|
| SERLIC | 0.03/0.03 | 0.09/0.13 | 0.06/0.07 | 0.09/0.13 | 0.18/0.23 |
| SERLIC (LIGHT) | 0.03/0.03 | 0.02/0.03 | 0.03/0.03 | 0.03/0.03 | 0.08/0.09 |

*Table 3.* Quantitative compression results of Attention-based and Mamba-based implementations

| CLASS | BPP | | CR GAIN |
| | ATTENTION-BASED | MAMBA-BASED | VS. ATTENTION-BASED |
|---|---|---|---|
| KITTI | 3.80 | 3.64 | -4.21% |

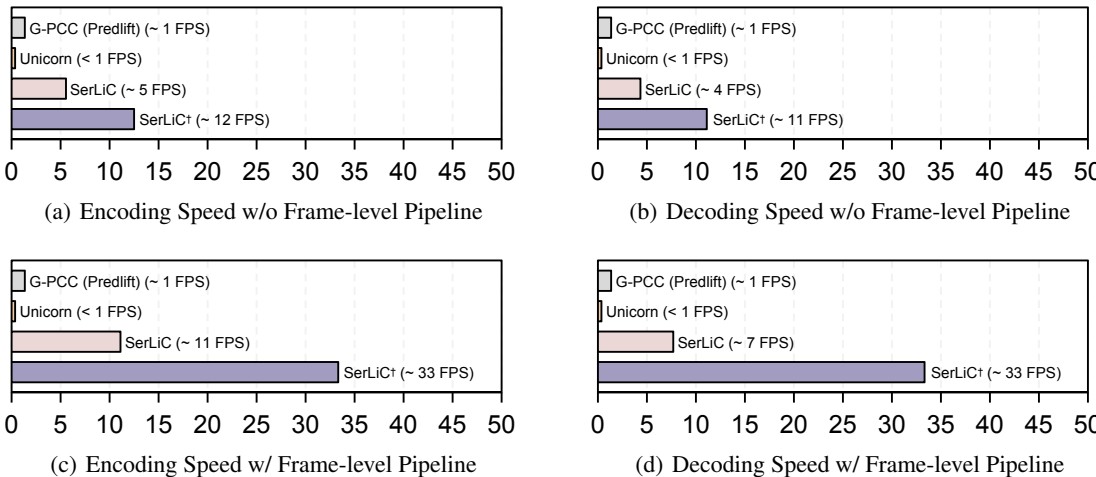

*Figure 3.* The coding speed of SerLiC w/o and w/ frame-level pipeline implementation.

*Table 4.* Computational complexity of Attention-based SerLiC at different window sizes. Maximum GPU memory (Mem.) and encoding/decoding time (seconds per frame) are analyzed. CC, NN, and AC refer to context construction, neural network, and arithmetic coding latencies, respectively.

| SIZE | MEM. (GB) | CC (S) | NN (S) | AC (S) | TOTAL(S) |
|------|-----------|--------|--------|--------|----------|
| 64   | 1.31      | 0.03/0.03 | 0.03/2.14 | 0.04/0.04 | 0.10/2.21 |
| 128  | 1.46      | 0.03/0.03 | 0.04/5.02 | 0.06/0.07 | 0.13/5.12 |
| 256  | 2.03      | 0.03/0.03 | 0.05/13.2 | 0.09/0.09 | 0.17/13.3 |
| 512  | 3.23      | 0.03/0.03 | 0.09/39.8 | 0.18/0.18 | 0.30/40.0 |
| 1024 | 5.84      | 0.03/0.03 | 0.15/137  | 0.32/0.33 | 0.50/138 |

## B.2. Attention Performance and Complexity

As described in the main manuscript, Attention can also be used in SerLiC for context modeling. However, compared to Attention, Mamba is more suited to our framework due to its inherent ability to capture point dependencies in a sequence.

To support our claim above, we provide a detailed analysis of both the coding performance and computational complexity when using Attention. This allows readers to better understand the comparative impact of Attention and Mamba on the overall system performance.

**Implementation Details.** Given the parallel window size $W$, the computational complexity of self-attention mechanism is $\Theta(W^2)$ due to the matrix-based multiplication computation between $Q$, $K$, and $V$ in self-attention. As we use the autoregressive coding method, a **causal mask** with the size of $W \times W$ (as indicated in Figure 3b of our main manuscript) is applied to ensure that each token in the sequence only uses its preceding tokens (its following tokens are unavailable as they are not coded yet) for context modeling. During the encoding process, since all points are known to the encoder, the model can process all tokens in parallel, and the complexity is accordingly $\Theta(W^2)$. However, during decoding, to generate an output sequence consistent with the encoding, the model must decode each point sequentially in a window $W$. As such, the attention mechanism results in a cubic time complexity $\Theta(W^3)$ in decoding.

By contrast, the autoregressive nature in Mamba naturally aligns with our SerLiC in the 1D point sequence. Its complexity is linear to the window size, i.e., $\Theta(W)$. Therefore, using Mamba is more efficient in our implementation.

**Coding Performance.** Table 3 reports the compression performance of Attention on the KITTI dataset when using the same window size (128) and parameter settings as Mamba. It is observed that Mamba even attains better results than Attention on KITTI, gaining 4.21% on average.

**Complexity.** Table 4 details the computational complexity of using the Attention block instead of Mamba in SerLiC. For fair comparisons, we list the runtime of sequential implementation (w/o frame pipelining). In comparison with Table 5 in our

main manuscript where the computational complexity of Mamba is presented, we observe that using Attention requires not only much higher GPU memory (1.46 GB vs. 0.41 GB when window size is 128) but also longer processing time (0.13/5.12 vs. 0.16/0.18 seconds per frame when window size is 128). Obviously, for the encoding time, Attention is comparable to Mamba. But for the decoding time, Attention is much longer, 28× of Mamba.

### B.3. L3C2 Analysis and Complexity

Limited by pages, the details of the L3C2 method are not provided in our main manuscript. In the following section, we will elaborate on L3C2 and its complexity for comparison with our SerLiC.

**L3C2 Introduction.** MPEG introduced L3C2 (Sébastien & Jonathan, 2021) in 2021, specifically tailored for the compression of LiDAR point clouds. However, its dependence on detailed sensor configurations (e.g., the precise pitch angles of individual lasers) limits its applicability on various datasets. Specifically, L3C2 requires the following inputs (refer to Figure 4):

(1) *numLasers*: the number of laser scan lines from the LiDAR;

(2) *lasersTheta*: the tangent of the pitch angle for each line, where negative values indicate a scan direction towards the ground, and positive values indicate a scan towards the sky;

(3) *lasersZ*: the distance from the laser head to the LiDAR center for each line, measured in millimeters;

(4) *lasersNumPhiPerTurn*: the number of points each line can scan in one full rotation.

For readers' convenience, we capture the configuration of L3C2 codec in Figure 4 to demonstrate the parameters required by L3C2. Since MPEG provides detailed LiDAR configuration parameters only for the Ford dataset, we are unable to test on KITTI and nuScenes in this work. Moreover, these parameters are only applicable to raw, unprocessed LiDAR point cloud data. However, the point cloud samples in nuScenes and KITTI datasets have undergone preprocessing before release. As a result, even if their LiDAR parameters could be derived, they would not be compatible with these preprocessed samples.

```
numLasers: 64
lasersTheta: -0.461611, -0.451281, -0.440090, -0.430000, -0.418945, -0.408667, -0.398

lasersZ: 29.900000, 26.600000, 28.300000, 24.600000, 26.800000, 25.100000, 24.800000,

lasersNumPhiPerTurn: 800,  800,  800,  800,  800,  800,  800,  800,  800,  800,  800,
```

*Figure 4.* Configuration of L3C2 codec, where four parameters are obtained from raw LiDAR physical parameters.

*Table 5.* Runtime comparison against L3C2 on the Ford dataset

| METHOD | ENC. | DEC. |
|---|---|---|
| L3C2 | 0.11s | 0.06s |
| SERLIC | 0.19s | 0.22s |
| SERLIC (LIGHT) | 0.08s | 0.08s |

**Complexity.** In Table 2 of our main manuscript, we compare the coding performance of SerLiC with that of L3C2. Furthermore, we compare their runtime in Table 5. It is observed that our SerLiC (light) runs slightly faster than L3C2 in encoding process, while slightly lower in decoding. Notice that L3C2 is implemented using C language and well optimized on CPU, whereas our SerLiC uses Python language and executes on both CPU and GPU. Thus, this comparison just serves as a reference for intuitive observation about the runtime of both methods.

## C. G-PCC Configuration

G-PCC offers an optional configuration, called the Angular mode, which includes detailed LiDAR parameters similar to those used in L3C2. Since these parameters are only available on the Ford dataset, we enable the Angular mode only when testing on the Ford dataset, and disable it for KITTI and nuScenes datasets.

*Table 6.* Detection results using the classical detection models w/ and w/o reflectance on the KITTI dataset

| METHODS | CAR | | | PED. | | | CYC. | | | MAP |
|---|---|---|---|---|---|---|---|---|---|---|
| | EASY | MOD. | HARD | EASY | MOD. | HARD | EASY | MOD. | HARD | MOD. |
| POINTPILLAR (W/ R) | 87.75 | 78.40 | 75.18 | 57.30 | 51.41 | 46.87 | 81.57 | 62.81 | 58.83 | **64.21** |
| POINTPILLAR (W/O R) | 83.85 | 74.21 | 70.36 | 21.56 | 14.08 | 12.83 | 47.79 | 34.28 | 32.13 | **40.86** |
| SECOND (W/ R) | 90.55 | 81.61 | 78.61 | 55.95 | 51.15 | 46.17 | 82.97 | 66.74 | 62.78 | **66.50** |
| SECOND (W/O R) | 87.87 | 78.92 | 75.58 | 40.46 | 35.66 | 31.88 | 70.42 | 50.64 | 47.70 | **55.07** |
| POINTRCNN (W/ R) | 91.47 | 80.54 | 78.05 | 62.96 | 55.04 | 48.56 | 89.17 | 70.89 | 65.64 | **68.82** |
| POINTRCNN (W/O R) | 88.03 | 77.00 | 72.85 | 42.90 | 35.28 | 30.68 | 56.58 | 42.51 | 40.07 | **51.60** |

## D. Downstream Tasks

In our introduction, we mention that "*...our experiments show that removing the reflectance causes a dramatic drop in pedestrian (and cyclist) detection accuracy incorporating the pre-trained PointPillar (Lang et al., 2019) detection model, having AP (average precision) from 51.4 (62.8) to 14.1 (34.3) on the KITTI dataset (see Figure 1)...*" To support our statement, we offer our full results on the KITTI dataset in Table 6.

The results in Table 6 provide a detailed comparison of the detection performance of three classical models—PointPillar, SECOND, and PointRCNN—with and without reflectance data. Note that our experiments are conducted on pre-trained models. The table shows the Average Precision (AP) for three categories: Car, Pedestrian (Ped.), and Cyclist (Cyc.), across different difficulty levels (Easy, Moderate, Hard), as well as the mean Average Precision (mAP) for the Moderate category.

It is observed that, reflectance data is essential for maintaining high accuracy and robustness in object detection tasks, particularly for pedestrians and cyclists. The substantial drop in AP for these categories underscores the importance of preserving reflectance information in point cloud data for downstream applications.

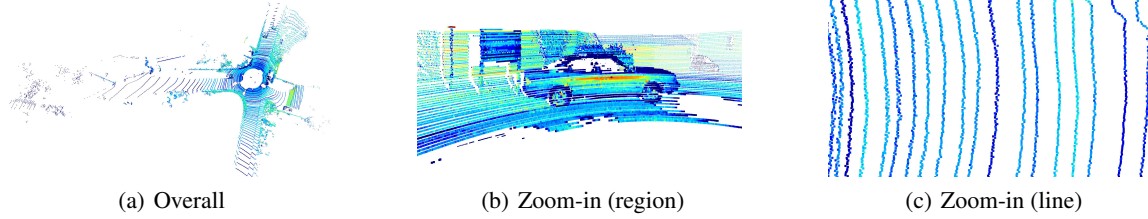

| (a) Overall | (b) Zoom-in (region) | (c) Zoom-in (line) |
|---|---|---|

*Figure 5.* Intuitive observation on reflectance characteristics in KITTI. (a) shows a holistic perspective. (b) shows zoom-in the region details. (c) further zooms in the reflectance of each sensor line for observation. The color indicates the value of reflectance, ranging from blue (low) to red (high).

*Table 7.* Ablation study on the sequence-level parallelization: exploiting correlations across sequences. "Intra" indicates intra-sequence coding, while "Inter (1)" and "Inter (3)" represent inter-sequence coding based on the preceding one or three sequences, respectively.

| | BPP | MEM. (GB) | NN (S) | TOTAL (S) |
|---|---|---|---|---|
| INTRA | 3.64 | 0.41 | 0.09/0.13 | 0.18/0.23 |
| INTER (1) | 3.64 | 0.04 | 5.04/5.82 | 5.20/9.84 |
| INTER (3) | 3.63 | 0.04 | 5.25/6.18 | 5.48/10.16 |

## E. More Ablation Studies

**Sequence Parallelization.** In Section 3.4—**Dual Parallelization**, we propose the dual parallelization scheme. including sequence level and window level, to accelerate coding speed of SerLiC within a frame. The ablation of window-level parallelization is presented in Section 4.5—**Window-based Parallelization**. Limited by page number, we depict the ablation study of **Sequence-level Parallelization** here.

On the other hand, in Section 3.4 – **Dual Parallelization**, we state that "*SerLiC processes point sequences in parallel, disregarding correlations across sequences. This is justified, as point sequences are scanned by separate laser sensor rotations, inherently lacking strong inter-sequence dependencies. As a result, sequence parallelization does not compromise coding efficiency but greatly enhances processing speed.*" Our ablation study here supports our statement above.

Figure 5 visualizes the reflectance of a point cloud frame. For better observation, we color the reflectance values using different colors. It is clear that reflectance values are close to each other in a scan line while different to some extent across scan lines. This intuitively reveals that using point correlations within a scan line leads to better performance. Furthermore, we conduct experiments which exploit correlations not only across points in a sequence but also across sequences for reflectance compression. In this way, the sequence-level parallelization is disabled. That is, we apply the autoregressive coding both across sequences (called *inter-sequence* coding in the following) and within a sequence (called *intra-sequence* coding in the following).

**Compression Performance.** Table 7 presents the experimental results. When leveraging correlations between two sequences, i.e., using the previous sequence for context modeling while encoding the current one, no gain is attained. The same holds when utilizing the previous three sequences. These findings is consistent with our observation that points are more strongly correlated within a single sequence. This further validates the rationale behind our sequence-level parallelization approach, which effectively maintains high coding performance while keeping the complexity low.

**Computational Complexity.** As the inter-sequence coding disable sequence-level parallelism, its memory cost is reduced. As reported in Table 7, the memory usage of inter-sequence coding is only 0.04 GB. However, the runtime is accordingly remarkably increased due to the autoregressive coding across sequences. For example, the total encoding/decoding time is 5.20/9.84 seconds per frame when using the previous sequence as context of the current sequence. When using three sequences, the coding time is even longer (5.48/10.16) as longer time is required by the context construction stage.

