# OpenReview forum: "Efficient LiDAR Reflectance Compression via Scanning Serialization"
_ICML.cc/2025/Conference — ICML 2025 poster_

### Official Review · Reviewer_wVmh · 2025-02-21

**Overall Recommendation:** 2

**Summary:**

This paper proposes a reflectance compression method based on serialized LiDAR data. Specifically, the method first converts 3D LiDAR point clouds into 1D sequences through scanning order serialization, where each point is labeled with a context representation that includes the sensor scan index, radial distance, and previous reflectance values, thereby establishing dependencies within the sequence. To achieve efficient sequence modeling, the paper combines Mamba with a two-level parallelization scheme to accelerate the autoregressive processing speed. Experimental results show that this method outperforms the latest GPCC and deep learning-based Unicorn in terms of model size, encoding/decoding time, and compression performance. Additionally, the authors propose a pipelined compression strategy, enabling the encoding speed to reach thirty frames per second, which has practical significance for real-world applications.

**Claims And Evidence:**

Yes.

**Essential References Not Discussed:**

None

**Experimental Designs Or Analyses:**

There are no significant issues with the experimental design and analysis.

**Methods And Evaluation Criteria:**

Yes.

**Other Comments Or Suggestions:**

None

**Other Strengths And Weaknesses:**

Strengths:
(1) The paper points out that the correlation of reflectance between different sequences may not be high, allowing for the parallel processing of point clouds from different sequences. This is a significant finding, as current research in both image compression and point cloud attribute compression has focused heavily on the similarity of local information (e.g., studies based on context-based entropy models). The discovery of the unique distribution pattern of LiDAR reflectance in point clouds provides substantial guidance for future research, particularly in real-time data encoding and decoding.
(2) The paper demonstrates potential for real-time encoding and decoding. The method employs parallelization between different sequences and within multiple windows of a single sequence, referencing the pipelining concept from computer architecture, transforming the encoding process into a three-stage pipeline, achieving a decoding speed of thirty frames per second.
(3) The paper includes comprehensive ablation experiments: (a) it compares the decoding time, model size, and runtime memory usage of the core module Mamba with that of Transformer; (b) the ablation experiments regarding the correlation of reflectance between different sequences are particularly convincing, as they visually illustrate the differences between sequences and experimentally demonstrate that referencing more sequences does not lead to performance improvements.
(4) The method proposed in this paper is specifically designed for LiDAR point clouds. Unlike the L3C2 method, which requires specific sensor parameter information, this method is more applicable to various datasets, needing only the most basic geometric information (xyz) to function.

Overall, the proposed method exhibits excellent compression performance and encoding/decoding speed. The writing of the paper is also quite reasonable, supported by sufficient experimental evidence to substantiate the authors' claims.

Weaknesses:
(1) The proposed reflectance is significant for downstream detection tasks. The experimental results indicate that the accuracy for bicycles and pedestrians drops to less than half. I would like to know whether the authors retrained the PointPillar model under the same settings without considering reflectance during this experiment. Additionally, I hope to see more comprehensive comparative results, such as detection accuracies for cars, pedestrians, and cyclists at different AP levels. Since the impact of reflectance on downstream tasks is one of the core points of this paper, I would like this section of the comparative experiments to be clearer and more thorough.
(2) In the process of data serialization, according to the formula provided by the authors in eq(4), there is an issue. When quantizing based on the elevation angle, due to noise during the sampling of LiDAR point clouds, it is inevitable that some different points will be mapped to the same (u, v) coordinates during the actual conversion process, which undoubtedly leads to a loss of precision. However, it seems that the paper claims to achieve lossless reflectance compression (as the authors do not provide RD curves in the subsequent results, only bpp and bpp-gain). I am concerned about the correctness of this part when compared to other results and hope the authors can provide a detailed explanation regarding this issue.
(3) The paper does not conduct experiments at multiple bitrate points, which raises some confusion. Other methods, such as G-PCC and Unicorn, provide results at multiple bitrate points. When comparing with these methods, did the authors set them all to lossless mode? Similarly, why did the authors not conduct tests at multiple bitrate points? Even with lossless compression, different quantizations of attributes can achieve various bitrate points.

**Questions For Authors:**

Please see the weakness.

**Relation To Broader Scientific Literature:**

I believe the main contribution of this paper is the proposal that the reflectance in different sequences of LiDAR may not be correlated. The authors have demonstrated this point through ablation experiments, which provides guidance for the parallelization of future LiDAR reflectance compression.

**Theoretical Claims:**

There are no apparent errors in the theoretical proof.

---

> ### Author Rebuttal · Authors · 2025-03-31
>
> We sincerely appreciate your insightful comments and constructive suggestions. Thank you for recognizing this paper *provides substantial guidance for future research* with *reasonable writing* and *sufficient experimental evidence*. Below, we provide detailed responses in the hope of addressing your concerns.
>
> **1. Clearer Comparative Experiments for Downstream Detection Task**
>
> 1) The visualizations and quantitative results in our manuscript show that simply removing reflectance data (setting values to 0) in pre-trained models leads to significant performance degradation. As suggested, expanded comparisons now include SECOND and PointRCNN (see table below), showing detection accuracies for cars/pedestrians/cyclists at multiple AP levels ("w/o R" = reflectance set to 0).
>
> ||Car Easy|Car Mod.|Car Hard|Ped. Easy|Ped. Mod.|Ped. Hard|Cyc. Easy|Cyc. Mod.|Cyc. Hard|**mAP Mod.**|
> |:-|:-:|:-:|:-:|:-:|:-:|:-:|:-:|:-:|:-:|:-:|
> |PointPillar |87.75| 78.40| 75.18| 57.30| 51.41| 46.87| 81.57| 62.81| 58.83| **64.21**|
> |PointPillar (w/o R)   |83.85| 74.21| 70.36| 21.56| 14.08| 12.83| 47.79| 34.28| 32.13| **40.86**|
> |SECOND |90.55| 81.61| 78.61| 55.95| 51.15| 46.17| 82.97| 66.74| 62.78| **66.50**|
> |SECOND (w/o R)   |87.87| 78.92| 75.58| 40.46| 35.66| 31.88| 70.42| 50.64| 47.70| **55.07**|
> |PointRCNN   |91.47| 80.54| 78.05| 62.96| 55.04| 48.56| 89.17| 70.89| 65.64| **68.82**|
> |PointRCNN (w/o R)|88.03| 77.00| 72.85| 42.90| 35.28| 30.68| 56.58| 42.51| 40.07| **51.60**|
>
> 2) We retrained the detection models from scratch without reflectance. The table below shows the performance of reflectance-ablated (*) models, where performance degradation is reduced but persists (and is non-trivial).
>
> ||Car Easy|Car Mod.|Car Hard|Ped. Easy|Ped. Mod.|Ped. Hard|Cyc. Easy|Cyc. Mod.|Cyc. Hard|**mAP Mod.**|
> |:-|:-:|:-:|:-:|:-:|:-:|:-:|:-:|:-:|:-:|:-:|
> |PointPillar*  | 87.86| 78.39| 75.71| 59.28| 52.48| 48.04| 76.18| 52.96| 50.17| **61.28**|
> |SECOND*  | 88.70| 79.47| 77.86| 57.52| 52.79| 48.38| 78.72| 56.92| 53.26| **63.06**|
> |PointRCNN*    | 89.46| 79.84| 77.58| 64.94| 54.62| 47.66| 89.16| 65.54| 60.92| **66.67**|
>
> 3) To more adequately reflect the role of reflectance, we revised our original wording from "It is crucial in downstream tasks..." to "It is widely used in downstream tasks...". All results will be detailed in the revised manuscript and supplementary.
>
> Finally, we respectfully emphasize that the core contribution of this work lies in the development of a high-performance real-time reflectance compressor. The extensive utilization of reflectance in downstream tasks provides substantial validation for the significance of our work.
>
> **2. Lossless Reflectance Compression vs. Quantization-Induced Precision Loss**
>
> We respectfully clarify that all comparisons were conducted in lossless mode, consistent with other lossless compression works (such as G-PCC and Unicorn).
>
> Although distinct coordinates may project to identical (u,v) pairs, these pairs only serve as auxiliary priors. The original 3D coordinates are preserved for each point through decoding the geometry bitstream (which aligns with G-PCC’s separate lossless geometry/attribute pipeline). Therefore, even though points may be mapped to the same (u,v) pair, they remain distinct points in 3D space, and we can clearly differentiate them based on their geomtry coordinates.
>
> Below we show an example. Consider two points in a point cloud, $p_1$=(3.12, 8.65, 0.50, 80) and $p_2$=(3.13, 8.63, 0.50, 82), where each point is represented as ($x$, $y$, $z$, $reflectance$). After coordinate transformation (in Eq. 3) and quantization (in Eq. 4), both points are mapped to the same (u,v) pair (712, 58). However, this spatial projection does not compromise the integrity of reflectance values. Specifically:
>
> - The encoder utilizes statistical patterns from the (712, 58) pair to optimize entropy coding for both values (80 and 82).
>
> - The decoder reconstructs reflectance values (80 and 82) by resolving the entropy-coded symbols using the same (u,v)-derived priors and the lossless geometry data.
>
> In short, SerLiC uses (u,v) as contextual priors for entropy coding while retaining raw reflectances. Please refer to the Python-style serialization code in our response to reviewer #Kwag for more details.
>
> **3. Additional Explanation on Bit Rate Points and Lossy Compression**
>
> Although different quantizations can adjust bitrates in lossless models, this approach differs fundamentally from lossy compression.
>
> We tested the quantization-driven lossy compression (suggested by the reviewer) against G-PCC, with RD curves available in [this anonymous website](https://anonymous.4open.science/r/11079-88B1/readme.md). It is observed that although SerLiC is designed for lossless compression, it remarkably outperforms lossy methods in the high bitrate range. Moreover, SerLiC operates in real-time, much faster than these lossy methods. Extending SerLiC to lossy compression is our future work.

---

### Official Review · Reviewer_Kwag · 2025-03-14

**Overall Recommendation:** 4

**Summary:**

The paper presents SerLiC, a novel serialization-based neural compression framework specifically designed for LiDAR reflectance data. The main contributions and findings of the study include:

1. Serialization of LiDAR Data: SerLiC transforms 3D LiDAR point clouds into 1D sequences through scan-order serialization, aligning with the LiDAR scanning mechanism. This approach allows for more efficient modeling and analysis of reflectance attributes.

2. Contextual Tokenization: Each LiDAR point is tokenized into a contextual representation that incorporates its sensor scanning index, radial distance, and prior reflectance values. This enhances the framework's ability to explore dependencies effectively.

3. Efficient Sequential Modeling: The framework utilizes the Mamba model, which operates with a dual parallelization scheme, facilitating simultaneous autoregressive dependency capture while ensuring fast processing.

4. Performance Improvements: SerLiC achieves over 2× volume reduction compared to the original reflectance data, with up to a 22% reduction in compressed bits relative to existing state-of-the-art methods like Unicorn. It operates with only 2% of the parameters used in these methods, making it highly efficient.

5. Real-world Applicability: A lightweight version of SerLiC, with 111K parameters, achieves processing speeds of ≥ 10 frames per second, making it practical for real-world applications such as autonomous driving and urban planning.

6. Experimental Results: Extensive tests demonstrate that SerLiC consistently outperforms existing compression methods (including G-PCC and Unicorn) across widely used datasets like KITTI, Ford, and nuScenes, reinforcing its effectiveness for LiDAR reflectance compression.

Overall, the paper highlights the need for specialized codecs tailored to LiDAR technology and presents SerLiC as a robust solution to meet this requirement.

**Claims And Evidence:**

(1) The paper introduces serialization to leverage the sequential nature of LiDAR scans, enabling more efficient modeling and processing. This approach is both logical and well-motivated.

(2) The paper thoroughly explores the intrinsic characteristics of LiDAR reflectance. Experiments demonstrate that removing reflectance leads to a significant decline in pedestrian and cyclist detection accuracy when using the widely adopted PointPillar model (Lang et al., 2019). Specifically, on the KITTI dataset, the average precision (AP) drops sharply from 51.4 (62.8) to 14.1 (34.3) for pedestrians (and cyclists), as shown in Figure 1, rendering it impractical for practical applications.

**Essential References Not Discussed:**

No.

**Experimental Designs Or Analyses:**

Yes, ablation experiment.

The ablation studies in the paper are well-designed, systematically evaluating the impact of contextual construction, window-based parallelization, the Mamba network, and attention mechanisms on the performance and computational complexity of the SerLiC model, thereby thoroughly validating the effectiveness of each component. In the contextual construction experiments, by individually disabling components such as scanning index, radial distance, and prior reflectance, the study clearly demonstrates their critical role in capturing LiDAR reflectance correlations. For instance, a noticeable performance drop occurs when the scanning index and radial distance are disabled, aligning with the physical characteristics of LiDAR scanning. The window-based parallelization experiments, by adjusting window sizes (e.g., from 64 to 1024), reveal a trade-off between performance and computational resources (e.g., encoding time and memory usage), identifying 128 as the optimal balance point, which reflects consideration for real-world deployment needs. The Mamba network experiments, varying the number of layers (1 to 7) and network dimensions (64 to 512), show that increased model capacity enhances performance but also raises complexity, a finding consistent with expectations and providing a basis for resource optimization. The attention mechanism experiments, by replacing the Mamba module, compare the performance and complexity of both architectures, proving that Mamba significantly reduces computational overhead (e.g., decoding time drops from 5.12 seconds to 0.23 seconds at a window size of 128) while maintaining comparable performance. Overall, these ablation studies comprehensively cover the model’s core components, employing a controlled variable approach to deliver clear causal insights. They not only confirm the rationality of the SerLiC design but also offer valuable guidance for practical applications in LiDAR reflectance compression. Future work could further explore cross-dataset generalization and the scalability of lossy compression to enhance the method’s applicability.

**Methods And Evaluation Criteria:**

The proposed method is both logical and well-motivated.

(1) Specifically, it introduces a lossless compression technique tailored for LiDAR point cloud reflectance data. Unlike previous approaches, which predominantly concentrate on general-purpose attributes such as color intensities or spatial coordinates, this method addresses a critical gap by focusing on the unique properties of LiDAR reflectance. General-purpose compression techniques, while versatile, often fail to fully exploit the specialized characteristics of reflectance data, such as its intensity distribution and correlation with surface materials. This limitation reduces their effectiveness in scenarios where reflectance plays a pivotal role, such as in autonomous driving, robotics, or environmental mapping. By prioritizing reflectance preservation without data loss, the proposed method offers a more efficient and targeted solution, enhancing the utility of LiDAR point clouds in these specialized applications.

The evaluation criteria are both reasonable and well-designed.

(1) The testing conditions adhere strictly to the Common Test Conditions (CTC) outlined by the MPEG AI-based Point Cloud Compression (AI-PCC) framework. This standardized approach ensures consistency and comparability with existing benchmarks in the field of point cloud compression. The CTC provides a rigorous set of guidelines, including predefined datasets, compression ratios, and performance metrics such as bitrate and reconstruction quality, allowing for an objective assessment of the method’s effectiveness. By aligning with these conditions, the evaluation not only validates the method’s performance under controlled and reproducible settings but also demonstrates its potential applicability within the broader context of AI-driven point cloud processing standards. This adherence strengthens the credibility of the results and facilitates future comparisons with other state-of-the-art techniques.

**Other Comments Or Suggestions:**

Discuss Limitations:
The paper should provide a deeper analysis of SerLiC’s limitations to enhance its credibility. For instance, its adaptability to non-rotational LiDAR systems, such as solid-state sensors with different scanning patterns, remains unaddressed. Additionally, the method’s performance under highly variable point cloud densities—common in complex environments like dense urban areas or sparse rural settings—needs exploration. These factors could affect serialization efficacy and compression quality. Discussing such constraints would clarify the method’s practical scope, guide future improvements, and help readers assess its applicability across diverse real-world scenarios.

**Other Strengths And Weaknesses:**

The concept of introducing serialization to leverage the sequential nature of LiDAR scans is insightful, enabling efficient modeling and processing. By transforming 3D LiDAR data into 1D sequences, it exploits inherent correlations, proving practical for real-world applications like autonomous driving. The writing is clear and accessible, effectively conveying complex ideas with concise explanations and figures, making it easy for readers, including non-specialists, to understand the methodology and its significance in LiDAR compression.

**Questions For Authors:**

【For Serialization Process:】
Could the authors provide a more detailed description or pseudocode for the serialization process to clarify how the LiDAR scanning order is transformed into a one-dimensional sequence?
A step-by-step explanation or pseudocode would help readers understand the exact mechanism behind converting the inherently spatial and temporal LiDAR scanning order into a 1D sequence, making the process more transparent and reproducible.
How does the serialization process specifically preserve the spatio-temporal correlations of LiDAR data? Please illustrate with a concrete example.
LiDAR data contains critical spatial and temporal relationships, and it’s unclear how these are maintained during serialization. A specific example—perhaps showing how a sequence of LiDAR points from a moving object retains its spatial continuity and temporal order—would make this preservation mechanism more tangible and convincing.

【For Comparison of Mamba Network and Transformer:】
Could the authors explain why the Mamba network is better suited for processing LiDAR reflectance data compared to Transformers? Please specify which characteristics of LiDAR data make Mamba more advantageous.
While both architectures are designed for sequence modeling, the authors should clarify why Mamba outperforms Transformers in this context. For instance, is it due to LiDAR data’s long-range dependencies, high sparsity, or variable sequence lengths? Highlighting these data-specific traits would strengthen the argument for choosing Mamba.
Are there experimental results or theoretical foundations that support the superiority of Mamba in long-sequence modeling? Please provide relevant data or literature references.
Claims of Mamba’s advantages need substantiation. Including experimental evidence (e.g., performance metrics like accuracy or efficiency on LiDAR datasets) or citing theoretical studies (e.g., prior work on Mamba’s efficiency in long-sequence tasks) would bolster the credibility of this comparison and provide a solid foundation for the authors’ conclusions.

**Relation To Broader Scientific Literature:**

The paper "Efficient LiDAR Reflectance Compression via Scanning Serialization" introduces SerLiC, advancing LiDAR point cloud compression by focusing on reflectance data, an underexplored area in prior work. Its key contributions—serializing point clouds by scanning order, using Transformer-based sequence modeling, implementing window-based parallelization, and integrating the Mamba network—build on and extend the scientific literature. Traditional methods, like those in Zhu et al. (2018) and de Queiroz and Chou (2016), prioritize spatial coordinates with general-purpose encoding, often neglecting reflectance-specific traits. SerLiC’s serialization leverages LiDAR’s sequential nature, akin to raster scans in image processing, enabling novel sequence modeling previously underutilized in this domain. The Transformer approach adapts established NLP techniques (e.g., Vaswani et al., 2017) to LiDAR data, while window-based parallelization ensures practical efficiency, a common deep learning practice. The Mamba network, a recent innovation (Gu & Dao, 2023), offers linear complexity, potentially marking its debut in LiDAR compression and outperforming prior methods like Unicorn (Wang et al., 2025) by achieving over 2x volume reduction and 22% bit rate improvement with minimal parameters. These advancements enhance compression efficiency and real-time applicability, contributing significantly to autonomous driving and 3D mapping research. Future work could explore broader generalization and lossy compression extensions.

**Theoretical Claims:**

There isn’t too much theoretical proof.

---

> ### Author Rebuttal · Authors · 2025-03-31
>
> **1. Deeper Analysis of SerLiC’s Limitations**
>
> We appreciate your insightful observations and are pleased to provide a comprehensive response below.
>
> 1.1 Adaptability to Non-Rotational LiDAR Systems.
>
> Non-rotational LiDAR's scanning also follows a specific order, and SerLiC maintains strong compatibility. To validate this, we have conducted additional experiments using the non-rotational InnovizQC dataset from MPEG. The dataset visualization is available in [this anonymous website](https://anonymous.4open.science/r/11079-88B1/readme.md). The compression results presented below demonstrate SerLiC's robust generalizability for non-rotational LiDAR systems.
>
> || RAHT | Predlift | SerLiC |
> |-:|:-:|:-:|:-:|
> | InnovizQC 02| 3.71 | 3.10 | 2.28 |
> | InnovizQC 03| 3.82 | 3.17 | 2.25 |
> | **Average** | **3.77** | **3.14** | **2.27** |
>
> 1.2 Performance Under Highly Variable Point Cloud Densities.
>
> We provide detailed performance across distinct scenarios in the table below. Results show that SerLiC consistently outperforms G-PCC in all scenes. Relatively, SerLiC performs better in City and Campus scenes than in Residential and Road scenes. This occurs due to dense roadside vegetation interfering with sensor-recorded reflectance characteristics and increasing compression difficulty. We will add these results to the supplementary to demonstrate the method's applicability across diverse real-world scenarios.
>
> || RAHT | Predlift | Ours |
> |-:|:-:|:-:|:-:|
> | City | 4.67 | 4.55 | 3.24 |
> | Residential | 4.96 | 4.94 | 3.81 |
> | Road | 5.41 | 5.36 | 4.39 |
> | Campus | 4.86 | 4.73 | 3.43 |
>
> **2. Pseudocode for the Serialization Process**
>
> The following Python-style code demonstrates the serialization of LiDAR point clouds. The function takes a point cloud of shape (N, 4) as input and outputs a list of point sequences, where each sequence corresponds to a specific laser emitter and represents an ordered point set scanned by that laser.
>
> ```python
> def scan_order_serialization(points, L=64, W=1024, pitch_up=3.0, pitch_down=-24.0):
>     """
>     Params:
>         points (torch.Tensor): Input point cloud of shape (N, 4), with columns (x, y, z, reflectance);
>         L (int): Number of lasers (vertical resolution);
>         W (int): Horizontal resolution;
>         pitch_up (float): Maximum elevation angle;
>         pitch_down (float): Minimum elevation angle;
>     Returns:
>         list[torch.Tensor]: List of point sequences. Each sequence corresponds to a laser beam, sorted by horizontal angle.
>     """
>     # Separate coordinates and reflectance values
>     coords, refl = points[:, :3], points[:, 3:4]
>
>     # Coordinate mapping (This step involves using Eq. 3 and 4
>     # from the manuscript to calculate rho, v, and u)
>     rho, v, u = coords_mapping(coords, L, W, pitch_up, pitch_down)
>
>     # Concatenate (rho, v, u) as auxiliary data
>     points = torch.cat([rho, v, u, coords, refl], axis=-1)
>
>     # Process each laser
>     seq_list = []
>     for laser_idx in range(1, L+1):
>         # Mask to filter points in the current laser
>         mask = (v == laser_idx).view(-1)
>         seq = points[mask]
>
>         # Sort points by horizontal index (u) to simulate spin order
>         spin_order = torch.argsort(u[mask].view(-1))
>         seq = seq[spin_order]
>         seq_list.append(seq)
>     return seq_list
> ```
>
> Our serialization approach transforms 3D LiDAR point clouds into 1D sequences for efficient modeling. Our serialization focuses on intra-frame organization, preserving each LiDAR frame's spatial structure losslessly. Temporal order is maintained by the system’s sequential storage/transmission of frames. For example, points from a moving car in frame $t$ retain their spatial layout after serialization; their temporal continuity across frames $t$, $t+1$, etc., is ensured by the ordered bitstream arrangement, unaffected by intra-frame encoding. This decoupling guarantees both spatial fidelity and temporal coherence.
>
> We will open-source our code to ensure transparency and reproducibility for the community.
>
> **3. Comparison of Mamba Network and Transformer**
>
> Specifically, the autoregressive coding employed in SerLiC is similar to the next token prediction in natural language processing (NLP). Autoregressive models are widely recognized for their exceptional contextual modeling capabilities, but their practical application is constrained by the high complexity. While Attention is also an alternative option for sequential modeling, our findings in Supplementary section C.2 show that Mamba exhibits significantly lower complexity-by several orders of magnitude-compared to attention-based architecture. This conclusion aligns with prior works [1]. Mamba’s linear complexity and dual-parallel strategy enable real-time processing of LiDAR reflectance compression. This is the key reason for our choice of Mamba.
>
> [1] Albert Gu, et al. "Mamba: Linear-Time Sequence Modeling with Selective State Spaces." Arxiv, 2023.

---

### Official Review · Reviewer_GMXH · 2025-03-15

**Overall Recommendation:** 2

**Summary:**

This paper proposes an algorithm for LiDAR point compression using LiDAR reflectance and serialization. While many studies only use the point location which is one part of the LiDAR sensor measures, this work focuses on the necessity of the LiDAR reflectance that may involve the surface attribute, which is also one important factor for the various downstream tasks, such as 3D object detection task.

Along with this concept, this paper extends to use the Mamba architecture for more efficiency and propose the entropy coding in Section 3.4.

The authors provide experimental results in terms of pointcloud compression in Table 1, and this method presents a plausible compression ratio compared to the other studies.

**Claims And Evidence:**

Overall, the authors well describe the importance of the LiDAR reflectance and the authors relate the LiDAR reflectance in the compression process using the Scan serialization. Serialization is recently used in the 3D scene understanding task, and this is also addressed in the Point Transformer v3 paper. In this perspective the authors well introduce this concept into the LiDAR point compression, so I believe that the overall claim is clear.

**Essential References Not Discussed:**

I believe that the authors should provide a technical comparison and experimental comparison with Point Transformer v3, which firstly provides a pointcloud serialization using conventional space carving algorithms. According to the authors' claims, I believe that this method can be utilized for not just 3D perception tasks, but also LiDAR compression using their serialization method. __If my understanding is correct, please provide the authors' analysis and comparison in the rebuttal.__

**Ethical Review Concerns:**

There is no ethical issue in this paper.

**Experimental Designs Or Analyses:**

In terms of the compression, the authors well provide the quantitative results which are included in Table 1. However, the authors did not provide the results of the 3D perception task which can be an important factor why the LiDAR reflectance is a necessary measurement. In the introduction section, the authors said

_"For instance, our experiments show that removing the reflectance causes a dramatic drop in pedestrian (and cyclist) detection accuracy incorporating the widely used PointPillar (Lang et al., 2019) detection model, having
AP (average precision) from 51.4 (62.8) to 14.1 (34.3) on the KITTI dataset (see Fig. 1), making it impractical for use."_

Without this, I cannot find any additional results in the manuscript and the supplementary material. Table 6 of the manuscript only provides results using PointPilar. Claiming the importance of the LiDAR reflectance solely from using the PointPillar results is not enough, in my opinion. It can mislead the readers.

Also, I am quite not sure why the Mamba architecture is a necessary condition for the LiDAR compression task. In my understanding, this formulation can be feasible when using Transformer architecture, like Point Transformer v3. In this perspective, the authors should provide a comparison with Scan Serialization with the conventional serialization methods. But, I cannot find the experimental results about this issue.

**Methods And Evaluation Criteria:**

While the claim itself is clear, I wonder about the logical relation between LiDAR point compression and Mamba architecture.

As the title said, this paper is about LiDAR point compression using LiDAR reflectance and the Scan serialization. Here, the Scan serialization is not the only property that Mamba architecture pursues. In my understanding, Transformer architecture can also encapsulate pointcloud as a 1D array, which is addressed by Point Transformer v3. Accordingly, the necessity of the Mamba architecture looks redundant and irrelevant to what the authors originally addressed in the title and the beginning of the introduction section.

In terms of evaluation criteria, the authors provide profound results in table 1. I can clearly see the compression ratio of this method and it achieves a promising result compared to results from others.

In the abstract and the beginning of the introduction section, the authors address the importance of LiDAR reflectance in the 3D perception task. Moreover, the teaser figure provides a failure case when the method does not use the LiDAR reflectance. However, I cannot find any quantitative experiments about this claim in the manuscript and the supplementary material. So, I would like to say that __some of the evaluation criteria are missing.__

**Other Comments Or Suggestions:**

I have no further comments. I wrote my questions and issues in the previous sections.

**Other Strengths And Weaknesses:**

The authors well describe the way of using the LiDAR reflectance for the LiDAR compression scheme. The scan serialization looks interesting since it respects the inherent properties that LiDAR sensors have. The proposed methodology looks okay and the resulting compression benefit is highly admirable as stated in Table 1.

However, my concern is closer to the necessity of introducing Mamba architecture into this compression task. With the provided writing in the introduction section, I cannot clearly catch why the authors should leverage the Mamba architecture to the task. I understand that the Mamba is more efficient than the Transformer architecture thanks to their linear computational complexity. However, such a viewpoint is more likely to say that the authors want to emphasize efficiency, rather than focusing on effective compression. I hope that the authors can describe their opinion of this issue in the rebuttal.

Another weakness is that the authors address the importance of using LiDAR reflectance in the 3D perception task, which is also stated in the previous section by the reviewer. Table 6 of the supplementary material only provides results using PointPilar. Except for this, I cannot be sure of the authors' claim.

Meanwhile, I am also confused about why the LiDAR compression is important to the 3D perception task itself. The experimental results imply that the authors put more weight on the efficiency. But, Table 6 and Line 53 are more about the performance itself. I am a bit confused about this. Can authors clarify this issue as well? I believe that if LiDAR reflectance itself is proven to be effective for LiDAR compression, the logic itself is good enough. What I wondered is why the statement about 3D perception is needed. I hope that I clarified my confusion to the authors.

**Questions For Authors:**

I have no further comments. I wrote my questions and issues in the previous sections.

**Relation To Broader Scientific Literature:**

This paper can be influential to future researchers who may use LiDAR sensors for their downstream tasks. As the authors mentioned, previous LiDAR-based perception methods mainly use the point location without using the LiDAR reflectance. Moreover, this paper also well designs the LiDAR point serialization while considering the LiDAR scan itself. So, I believe it is impactful to this field.

**Theoretical Claims:**

There are no theoretical claims that the authors provide. The authors revisit the concept of the serialization and the LiDAR reflectance. I believe that this paper is more closer to the technical paper, rather than to the theoretical paper.

__So, I wonder whether this paper deserves to be reviewed as an ICML submission paper.__ If this paper is submitted to other computer vision conferences, it is okay for me. But I will __not__ consider this issue in my rating and will check the feedback from AC or PC.

---

> ### Author Rebuttal · Authors · 2025-03-31
>
> We sincerely appreciate your thoughtful feedback. Thank you for recognizing the *highly admirable compression benefit* presented in our study and acknowledging its potential to be *influential to future researchers* and *impactful to this field*. We highly value this opportunity to address your concerns.
>
> **1. Relation Between LiDAR Point Cloud Compression, LiDAR Reflectance Compression, and 3D Perception Task**
>
> **LiDAR point cloud compression** reduces data volume from 3D LiDAR scans, which consist of spatial coordinates (geometry) and reflectance values (attributes), each requiring distinct compression methods:
>
> - **LiDAR geometry compression** encodes coordinates into a compact bitstream.
> - **LiDAR reflectance compression** is dedicated to optimizing attribute data into another bitstream, using geometry as conditional prior.
>
> The divergent statistical properties of geometric and reflectance data have historically motivated their independent compression solutions. This paper focuses on the **LiDAR reflectance compression**, significantly improving compression efficiency for reflectance data.
>
> Existing research predominantly advances LiDAR reflectance compression through technical refinements, while overlooking a fundamental systems-level inquiry: *"Is reflectance compression necessary, and how does it impact downstream tasks?"* We first address this gap through task-driven analysis, as illustrated in the introduction section and supplementary. Our evidence confirms that reflectance data significantly influences perception performance, thereby validating its compression and transmission necessity. Having established this critical premise, we subsequently propose the coding paradigm. Our contribution not only advances compression technology but also formally links reflectance fidelity to perception needs.
>
> **2. Logical Relation Between LiDAR Reflectance Compression and Mamba Architecture**
>
> We sincerely thank the reviewer for the constructive feedback. We would like to clarify that in the compression domain, efficiency and performance are equally critical considerations. Specifically, the autoregressive coding employed in SerLiC is similar to the next token prediction in natural language processing (NLP). Autoregressive models are widely recognized for their exceptional contextual modeling capabilities [1][2], but their practical application has been constrained by the high computational complexity. While Mamba is not the only option for sequential modeling, our findings in Supplementary section C.2 demonstrate that Mamba exhibits much lower complexity-by several orders of magnitude-compared to attention-based architectures under equivalent layer stacking and channel dimensions. By combining Mamba’s linear complexity with our proposed dual-parallel strategy, SerLiC achieves real-time processing capabilities. This is the key reason for our choice of Mamba.
>
> **3. Quantitative Experiments for the Importance of the LiDAR Reflectance**
>
> Following the reviewer's suggestion, we have conducted additional experiments using another two classical models (SECOND and PointRCNN), with the experimental results presented in the table below ("w/o R" denotes the performance of the complete removal of reflectance information).
>
> ||Car|Pedestrian|Cyclist|**mAP**|
> |:-|:-:|:-:|:-:|:-:|
> |PointPillar| 78.40 | 51.41 | 62.81 | **64.21** |
> |PointPillar (w/o R)| 74.21 | 14.08 | 34.28 | **40.86** |
> |SECOND| 81.61 | 51.15 | 66.74 | **66.50** |
> |SECOND (w/o R)| 78.92 | 35.66 | 50.64 | **55.07** |
> |PointRCNN| 80.54 | 55.04 | 70.89 | **68.82** |
> |PointRCNN (w/o R)| 77.00 | 35.28 | 42.51 | **51.60** |
>
> Nonetheless, we respectfully clarify that the core contribution of our work lies in the development of a high-performance real-time reflectance compressor. The extensive utilization of reflectance in downstream tasks provides substantial validation for the significance of our work.
>
> **4. Compression Using Serialization Methods in Point Transformer v3**
>
> While Point Transformer v3 (PTv3) is designed for perception tasks and its shift-based patch interaction isn’t directly applicable to compression, its serialization method offers an alternative to our scan-order approach. Following the reviewer's suggestion, We conduct experiments in SemanticKITTI. Results show that Hilbert Curve (3.74 Bpp) and Z-order Curve (3.70 Bpp) underperform our scan-order serialization (3.64 Bpp), confirming the efficacy of our method. The detail table is available in [this anonymous website](https://anonymous.4open.science/r/11079-88B1/readme.md).
>
> [1] David Minnen, et al. "Joint Autoregressive and Hierarchical Priors for Learned Image Compression." NeurIPS, 2018. \
> [2] Chunyang Fu, et al. "OctAttention: Octree-based large-scale contexts model for point cloud compression." AAAI, 2022.

---

> > ### Comment · Reviewer_GMXH · 2025-04-08
> >
> > I thank the authors for the precise rebuttals. Overall, my concerns are relieved and I have no further questions about the manuscript as well as the rebuttal. Among the answers, the explanation of using Mamba looks okay and I respect the authors' design choice.
> >
> > In short, I raise my score to __weak accept__. Thank you for the rebuttal and your endeavors.
> >
> > Best,

---

### Decision · Program_Chairs · 2025-05-01

**Decision:**

Accept (poster)

**Comment:**

This paper received divergent scores after the rebuttal. Reviewer Kwag maintained Accept, and  GMXH promised to raise to Weak Accept rating in discussion, while Reviewer wVmh remained at Weak Reject.

Reviewers GMXH and Kwag praised the paper for its novel approach to LiDAR reflectance compression via scan-order serialization and the incorporation of the Mamba architecture. They liked the balance between efficiency and performance for real-time compression, as well as the extensive quantitative results and ablation studies. There are also concerns raised regarding the theoretical motivation for introducing the Mamba architecture into the compression task, as well as the comprehensive evaluation of the impact on downstream tasks. Despite these issues, the rebuttal largely addressed the concerns.

After carefully reading the paper, reviews, and rebuttal, the AC concludes that although the remaining concerns are valid, the paper demonstrates sufficient technical novelty and strong empirical performance. Based on these considerations, the AC would like to recommend acceptance, with the expectation that the authors will address the comments in the final version.